# Evaluation of FluSight influenza forecasting in the 2021–22 and 2022–23 seasons with a new target laboratory-confirmed influenza hospitalizations

Accurate forecasts can enable more effective public health responses during seasonal influenza epidemics. For the 2021–22 and 2022–23 influenza seasons, 26 forecasting teams provided national and jurisdiction-specific probabilistic predictions of weekly confirmed influenza hospital admissions for one-to-four weeks ahead. Forecast skill is evaluated using the Weighted Interval Score (WIS), relative WIS, and coverage. Six out of 23 models outperform the baseline model across forecast weeks and locations in 2021–22 and 12 out of 18 models in 2022–23. Averaging across all forecast targets, the FluSight ensemble is the 2nd most accurate model measured by WIS in 2021–22 and the 5th most accurate in the 2022–23 season. Forecast skill and 95% coverage for the FluSight ensemble and most component models degrade over longer forecast horizons. In this work we demonstrate that while the FluSight ensemble was a robust predictor, even ensembles face challenges during periods of rapid change.

Traditional influenza surveillance systems provide a comprehensive picture of influenza activity in the United States[1–3] and are fundamental for situational awareness and risk communication. However, they measure influenza activity after it has occurred, and do not directly anticipate future trends to inform risk assessment and healthcare preparedness. To address these limitations, the Centers for Disease Control and Prevention (CDC) has supported open influenza forecasting challenges since the 2013–14 season[4]. This collaborative process (named FluSight) has ensured that forecasting targets are relevant to public health. Additionally, forecast data are openly available, which enables transparent evaluation of forecast performance[5,6].

Originally the FluSight collaboration focused on short-term forecasts of outpatient influenza-like-illness (ILI) rates from ILINet[2] and corresponding results have been summarized previously[4–6]. However, the COVID-19 pandemic resulted in changes in outpatient care-seeking behavior, and the continued co-circulation of SARS-CoV-2 has further complicated the interpretation of ILI data. In the 2021–22 influenza season, the FluSight forecast target shifted to the weekly number of hospital patients admitted with laboratory-confirmed influenza from the Health and Human Services (HHS) Patient Impact and Hospital Capacity Data System[7]. This system was created during the COVID-19 pandemic to gather a complete and unified representation of COVID-19 disease outcomes along with other metrics related to healthcare capacity. Hospitals registered with the Centers for Medicare and Medicaid Services (CMS) are required to report daily COVID-19 and influenza information[8]. Reporting of the influenza data elements, including the previous day's number of admissions with laboratory-confirmed influenza virus infection, became mandatory on February 2, 2022,[8]. Although influenza activity, has been monitored throughout the US for decades through multiple surveillance systems, this dataset is the first with laboratory-confirmed influenza hospital admissions reported systematically across all 50 states and additional territories[1–3,8].

The COVID-19 pandemic disrupted the typical timing, intensity, and duration of seasonal influenza activity in the United States and many parts of the world[9,10]. Influenza activity was very low during the

e-mail: nqr2@cdc.gov; xhq2@cdc.gov

2020–21 season in the U.S., but activity increased during the 2021–22 season, with activity peaking later in April, May, and early June 2022 and remaining at higher levels than had been reported during these months in previous seasons[10]. In the 2022-23 influenza season, activity began increasing nationally in early October, earlier than previous seasons[2,3,11], and peaked in early December 2022.

In this analysis, we summarize the accuracy and reliability of ensemble and component 1- to 4-week ahead forecasts of laboratory-confirmed influenza hospital admissions submitted in real-time during the 2021–22 and 2022-23 seasons. Our objective was to consider potential changes in performance of these forecasts in post-COVID influenza seasons, especially given atypical timing and intensity. By evaluating forecast performance for a new forecast target with limited calibration data, we identify specific areas for forecast improvement.

## Results

The 2021–22 influenza season was characterized by two distinct waves of activity. The first occurred between November 2021 and January 2022 and the second between February and June 2022, though reporting of influenza hospitalizations was not mandatory in the HHS system until February 2, 2022 (see observed data in Fig. 1a). Reported national weekly influenza hospital admissions exceeded 1000 for 22 out of 25 of the forecast weeks (Fig. 1a). Updates to weekly counts from the forecast evaluation period were generally minimal (Supplementary Figs. 2–4), with 94% of updates during the 2021–22 season resulting in

changes of under 10 hospitalizations for subnational jurisdictions. There were infrequent larger updates (10 or greater) to reported admissions.

The 2022-23 influenza season was characterized by an early start, reaching 1000 hospital admissions nationally before October 2022. A sharp increase nationally through October and November led to a peak of 26,600 hospital admissions in early December. Hospital admissions decreased rapidly after December, with 3000 weekly hospital admissions by the end of January, and eventually dropped below 1000 confirmed weekly admissions nationally by May 2023. Weekly numbers of admissions exceeded 1000 for 27 out of 34 of the forecast weeks (Fig. 1b, Supplementary Fig. 4). In the 2022–23 season, 83% of updates for weekly admissions resulted in changes of under 10 hospitalizations for subnational jurisdictions. There were infrequent larger updates to reported admissions and often updates occurred within two weeks of initial publication.

### Models Included

For both the 2021–22 and 2022–23 influenza seasons, 26 modeling teams submitted forecasts, and 21 and 16, respectively, were eligible for end-of-season evaluation, not including the FluSight baseline and ensemble models. The number and types of models included in the primary analysis (based on the inclusion criteria) varied across weeks with a range of methodological approaches (see Supplementary Table 1). For the 2021–22 season, a median of 20 included models was

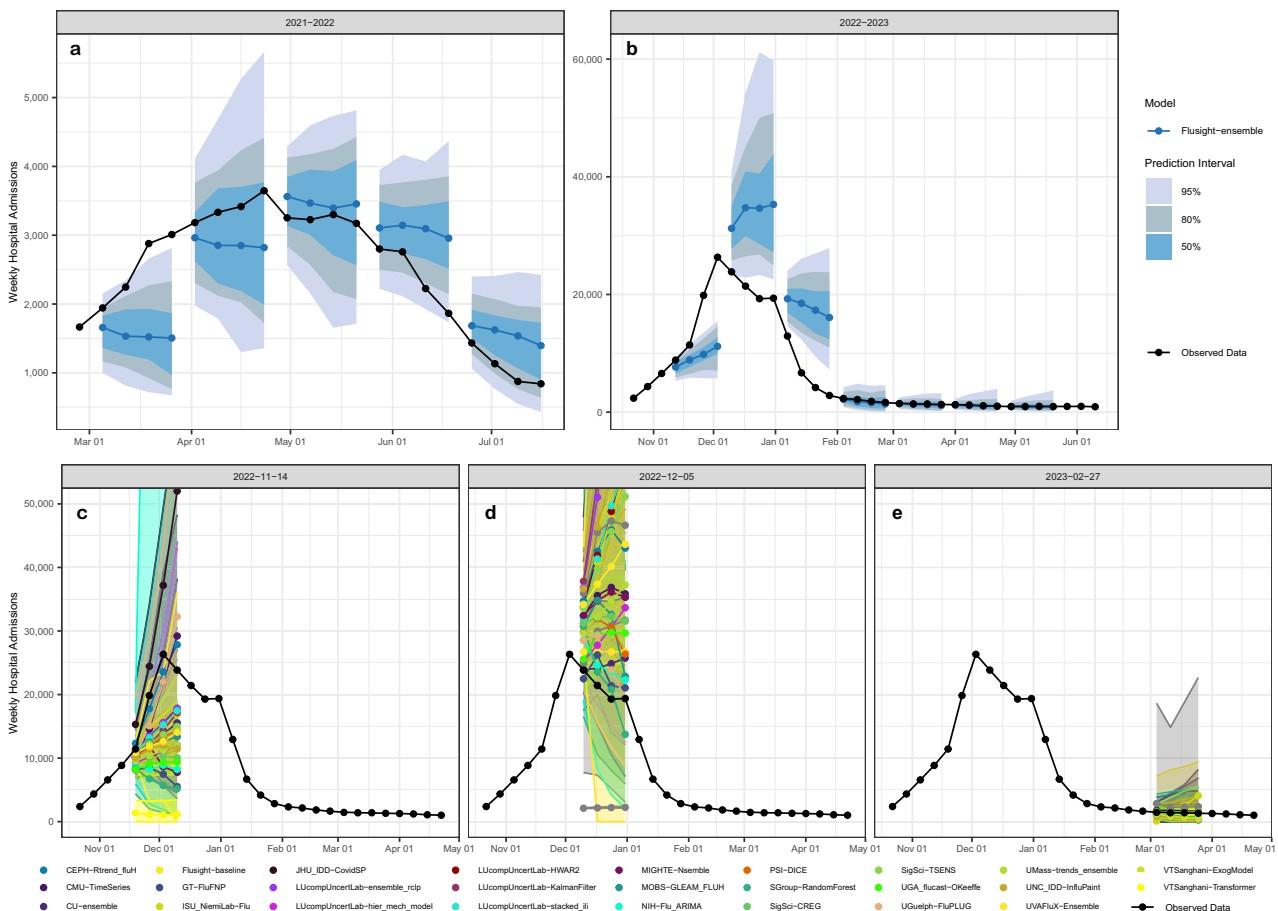

**Fig. 1 | National incident weekly hospital admissions and select forecasts.** National weekly observed hospitalizations (black points) along with FluSight ensemble forecasts for four weeks of submissions in the 2021–22 season (**a**) and seven weeks of submissions in the 2022-23 season (**b**). The median FluSight ensemble forecast values (blue points) are shown with the corresponding 50%, 80%, and 95% prediction intervals (blue shaded regions). **c–e** Show national

incident weekly hospital admissions (black points) from the 2022-23 season and predictions from all models submitted on November 11, 2022 (**c**), December 05, 2022 (**d**) and February 27, 2023 (**e**). Colored bands indicate 95% prediction intervals for each model. Team forecasts for additional weeks are available in an interactive dashboard[12].

submitted (range: 15–21), with most having a statistical component, three mechanistic, and six ensembles of component models. In 2022–23 there was a median of 15 included models (range: 10–16) submitted each week, with many having a statistical component, three mechanistic, and four ensemble models. Top performing models in the 2021–22 season included statistical, mechanistic and ensemble models. In 2022–23, top performing models included mechanistic, statistical, ensemble, and one machine learning model. There were also statistical, mechanistic, AI or machine learning, and ensemble models among models with lower performance across seasons. Modeling teams varied across seasons, with 13 modeling groups having submitted eligible forecasts for both seasons. When only national forecasting targets were considered, no additional teams were included for the 2021–22 season, but two teams, NIH-Flu_ARIMA and ISU_NiemiLab-Flu, met the inclusion criteria for 2022–23 (Supplementary Analysis 3). Visualizations of all forecasts as of the date they were submitted are included in an interactive dashboard[12].

### Relative WIS

Over the evaluation period, more models outperformed the FluSight baseline model in 2022–23 (12) than in 2021–22 (6) based on relative WIS (Table 1). Within each season, the models that achieved an overall relative WIS less than or equal to one represent a variety of modeling strategies, including a basic quantile autoregression fit, a mechanistic compartmental model with stochastic simulations, an ensemble of time-series baseline models, a random walk model, a random forest ensemble, and the FluSight ensemble (Supplementary Table 1). Similar results were observed when models were evaluated based on absolute error of the median of probabilistic forecasts (see MAE estimates in Table 1).

Few teams outperformed the FluSight Ensemble in relative WIS for both seasons. The CMU-TimeSeries model was the only model that outperformed the ensemble for both the 2021–22 and 2022–23 seasons, while the MOBS-GLEAM_FLUH, PSI-DICE, and MIGHTE-Nsemble models outperformed the ensemble only in the 2022–23 season.

For both seasons, forecasts from the FluSight Ensemble were ranked among the top 50% of all model forecasts for the same location, date, and target, more than three-fourths of the time (79.73% in 2021–22 and 78.83% in 2022–23) (Fig. 2). Three models consistently ranked in the top 25% for 2021–22 and 2022–23 seasons, respectively: CMU-TimeSeries (42.47%, 36.14%), PSI-DICE (39.34%, 39.87%), and MOBS-GLEAM_FLUH (38.97%, 50.33%). Several models, seven in 2021–22 and five in 2022–23, had bimodal rank distributions, with a combined majority of their forecasts falling in either the bottom 25% or top 25% (Fig. 2).

### Log-transformed analysis

For both seasons, the analysis using log-transformed hospitalization counts resulted in the same top five performing teams in terms of absolute and relative WIS. For the 2021–22 season, all teams were ranked the same for the log-transformed and non-transformed analyses. In 2022–23, MIGHTE-Nsemble and PSI-DICE performed better than CMU-TimeSeries for the log-transformed analysis (Table 1 and Supplementary Analysis 2).

### Relative WIS and Spatial Variation

Model performance varied by spatial jurisdiction. For individual states, relative WIS values varied across models ranging from 0.46 to 12.58 in 2021–22 and 0.32 to 12.35 in 2022–23 (Fig. 3). More models, including the ensemble, performed better at the state-level than the baseline in 2022–23 compared to 2021–22. The relative WIS of the FluSight Ensemble had the smallest range of values across all locations from 0.58 to 1.08 in 2021–22 to 0.63 to 1 in 2022–23 (Fig. 3 and Supplementary Fig. 1). To further examine forecast performance across jurisdictions, we considered the percent of jurisdictions that the relative

WIS value for a given model and location pair was less than the baseline (i.e., lower than 1). The FluSight Ensemble performed as well as or better than the baseline for all forecast jurisdictions for 2022–23 and 47 out of 52 forecast jurisdictions for 2021–22, a larger number of jurisdictions than any submitted model (Fig. 3). In 2022–23, 12 models performed better than the baseline at the jurisdiction-level at least 50% of the time, compared to five models in 2021–22. In general, the models with lower (better) relative WIS values were consistent between the analysis with all spatial jurisdictions and the analysis considering only national forecast targets for both seasons (Supplementary Analysis 3).

### Absolute WIS

Across forecasted weeks, the FluSight Ensemble's worst performance in terms of absolute WIS (maximum values) for 1-week ahead targets occurred on March 19, 2022 for 2021–22 and on November 26, 2022 for 2022–23 (Fig. 4). For the 4-week ahead horizon, maximum absolute values, indicating the worst performance, for each season occurred on June 04, 2022, and December 03, 2022, respectively (Fig. 4). Minimum, or best, absolute WIS values for each season occurred on July 16, 2022, and May 13, 2023, respectively, both during periods of low flu activity.

### Coverage

Model performance for the FluSight Ensemble dropped during periods of relatively rapid change (see Figs. 1 and 3). The lowest 1-week horizon 95% value occurred for forecasts with target end dates of March 14, 2022, for 2021–22 and on November 21, 2022, for 2022–23 (Fig. 5). Across forecasted weeks in the 2021–22 season, the FluSight Ensemble had a minimum 95% coverage value at the 1-week horizon of 75%. Lower 95% coverage for the 1-week horizon was observed in the 2022–23 season with a minimum of 29%. The maximum coverage rate achieved by the FluSight Ensemble in any individual week was 100% in both seasons. Minimum FluSight Ensemble 95% coverage values for forecasts at the 4-week horizon in any individual week were 62% for 2021–22 and 15% for 2022–23.

Model performance, in terms of coverage, tended to decline at longer time horizons for the FluSight Ensemble, baseline, and individual contributed models (see Table 2). Over the forecast weeks, the 2021–22 FluSight ensemble had slightly higher overall 95% coverage values of 89.32%, 86.11%, 85.15%, and 83.33% for the 1 to 4-week ahead horizons respectively, compared to the 2022–23 season during which the FluSight Ensemble had 95% coverage values of 85.79%, 81.64%, 78.78%, and 77.85% for the 1 to 4-week ahead horizons respectively. A similar proportion of models had higher overall 95% coverage values at the 1-week ahead horizon than at the 4-week ahead horizon for 2022–23 (14 of 18 models) and 2021–22 (18 of 23 models) (Table 2). Out of the forecast targets and across forecast weeks, the FluSight Ensemble's 95% prediction interval contained at least 90% of the corresponding observed values only 55.56% and 64.52% of the time, for 2021–22 and 2022–23, respectively (Table 2). Ideally 95% prediction intervals are just wide enough to capture 95% of eventually observed values.

## Discussion

The 2021–22 influenza season marked the return of from very low levels of seasonal influenza activity observed in the U.S. following the first years of the COVID-19 pandemic, and many components of the 2021–22 and 2022–23 FluSight Forecasting Challenges were new. One of the most substantial changes was the shift from the original FluSight forecasting targets of weekly influenza-like-illness (ILI) percentages to weekly counts of confirmed influenza hospitalizations. The COVID-19 pandemic resulted in the availability of a new data source, the unified HHS-Protect dataset, which provided information on laboratory-confirmed daily influenza hospitalizations from all 50 states, D.C., and Puerto Rico[7,8]. Confirmed influenza hospital admissions may more

**Table 1 | Performance metrics for teams meeting inclusion criteria**

| Model | Absolute WIS | Relative WIS | MAE | 50% Coverage (%) | 95% Coverage (%) | % of Forecasts Submitted | Log Absolute WIS | Log Relative WIS |
|---|---|---|---|---|---|---|---|---|
| **2021-22** | | | | | | | | |
| CMU-TimeSeries[STAT] | 12.54 | 0.74 | 18.92 | 47 | 90 | 100 | 0.31 | 0.79 |
| Flusight-ensemble[ENS] | 13.86 | 0.82 | 20.79 | 48 | 86 | 100 | 0.33 | 0.83 |
| PSI-DICE[MECH] | 14.03 | 0.83 | 20.17 | 43 | 82 | 100 | 0.33 | 0.84 |
| UMass-trends_ensemble[ENS] | 14.35 | 0.85 | 22.24 | 71 | 97 | 100 | 0.36 | 0.91 |
| SGroup-RandomForest[ENS] | 15.45 | 0.91 | 23.87 | 47 | 95 | 100 | 0.38 | 0.97 |
| CEID-Walk[STAT] | 15.63 | 0.93 | 22.19 | 52 | 82 | 89 | 0.39 | 0.98 |
| Flusight-baseline[STAT] | 16.99 | 1.00 | 24.10 | 49 | 83 | 100 | 0.40 | 1.00 |
| MOBS-GLEAM_FLUH[MECH] | 17.17 | 1.02 | 22.25 | 0.32 | 0.63 | 91 | 0.42 | 1.07 |
| GT-FluFNP[STAT] | 17.57 | 1.03 | 23.40 | 0.39 | 0.69 | 96 | 0.38 | 0.98 |
| SigSci-TSENS[ENS] | 17.79 | 1.03 | 24.86 | 38 | 72 | 96 | 0.40 | 1.01 |
| IEM_Health-FluProject[STAT] | 17.69 | 1.05 | 23.98 | 50 | 85 | 100 | 0.40 | 1.02 |
| CU-ensemble[ENS] | 18.32 | 1.08 | 25.41 | 44 | 77 | 100 | 0.39 | 0.98 |
| LUcompUncertLab-TEVA[ENS,STAT] | 21.02 | 1.20 | 29.99 | 54 | 86 | 89 | 0.41 | 1.04 |
| UVAFluX-Ensemble[ENS] | 21.65 | 1.27 | 25.76 | 38 | 64 | 99 | 0.45 | 1.14 |
| LUcompUncertLab-VAR2_plusCOVID[STAT] | 22.03 | 1.30 | 28.99 | 42 | 74 | 94 | 0.42 | 1.08 |
| LUcompUncertLab-VAR2K_plusCOVID[STAT] | 24.44 | 1.39 | 32.43 | 0.42 | 0.74 | 85.19 | 0.47 | 1.19 |
| UT_FluCast-Voltaire[STAT] | 23.64 | 1.39 | 35.19 | 0.50 | 0.91 | 95.13 | 0.45 | 1.15 |
| LUcompUncertLab-VAR2[STAT] | 25.93 | 1.53 | 35.05 | 39 | 72 | 94 | 0.53 | 1.35 |
| LUcompUncertLab-VAR2K[STAT] | 26.81 | 1.54 | 39.35 | 42 | 83 | 89 | 0.61 | 1.54 |
| LosAlamos_NAU-CModel_Flu[STAT, MECH] | 28.69 | 1.70 | 36.14 | 26 | 59 | 100 | 0.63 | 1.62 |
| SGroup-SIkJalpha[STAT] | 28.94 | 1.70 | 38.59 | 18 | 46 | 100 | 0.49 | 1.24 |
| GH-Flusight[ENS] | 30.93 | 1.81 | 31.89 | 6 | 13 | 94 | 0.74 | 1.88 |
| SigSci-CREG[STAT] | 27.36 | 1.97 | 31.00 | 19 | 44 | 89 | 0.80 | 2.06 |
| **2022-23** | | | | | | | | |
| MOBS-GLEAM_FLUH[MECH] | 42.20 | 0.61 | 57.97 | 42 | 78 | 94 | 0.37 | 0.66 |
| CMU-TimeSeries[STAT] | 44.48 | 0.67 | 65.94 | 49 | 87 | 94 | 0.41 | 0.70 |
| PSI-DICE[MECH] | 47.45 | 0.70 | 63.17 | 48 | 80 | 100 | 0.40 | 0.70 |
| MIGHTE-Nsemble[ENS,AI/ML, STAT] | 48.99 | 0.73 | 67.50 | 53 | 82 | 96 | 0.41 | 0.70 |
| Flusight-ensemble[ENS] | 51.72 | 0.77 | 71.04 | 56 | 81 | 100 | 0.44 | 0.74 |
| UMass-trends_ensemble[ENS] | 53.86 | 0.80 | 79.40 | 63 | 89 | 100 | 0.49 | 0.83 |
| GT-FluFNP[STAT] | 59.75 | 0.81 | 72.88 | 56 | 75 | 89 | | 0.90 |
| SGroup-RandomForest[ENS] | 54.29 | 0.82 | 75.98 | 53 | 84 | 97 | 0.52 | 0.87 |
| CU-ensemble[ENS] | 62.23 | 0.83 | 75.57 | 51 | 70 | 84 | 0.51 | 0.85 |
| CEPH-Rtrend_fluH[STAT] | 54.20 | 0.84 | 70.47 | 44 | 78 | 86.87 | 0.58 | 1.07 |
| UGA_flucast-OKeeffe[STAT] | 62.13 | 0.93 | 77.33 | 50 | 72 | 91 | 0.61 | 1.02 |
| VTSanghani-ExogModel[AI/ML] | 72.30 | 0.98 | 92.56 | 30 | 61 | 80 | 0.63 | 1.04 |
| Flusight-baseline[STAT] | 67.69 | 1.00 | 80.05 | 49 | 74 | 100 | 0.59 | 1.00 |
| SigSci-TSENS[ENS] | 64.27 | 1.00 | 80.02 | 58 | 74 | 93 | 0.66 | 1.11 |
| UNC_IDD-InfluPaint[STAT] | 61.14 | 1.05 | 77.90 | 40 | 75 | 76 | 0.52 | 0.96 |
| UVAFluX-Ensemble[ENS] | 78.71 | 1.11 | 94.45 | 22 | 41 | 95 | 0.61 | 1.02 |
| SigSci-CREG[STAT] | 79.68 | 1.33 | 89.29 | 38 | 62 | 91 | 0.68 | 1.16 |
| JHU_IDD-CovidSP[MECH] | 129.16 | 1.88 | 174.98 | 48 | 80 | 81 | 0.49 | 0.82 |

Forecast metrics are across all fifty states, D.C., and Puerto Rico forecast targets. The season is indicated in bold in the model column. The Absolute WIS column refers to the Weighted Interval Score for each model. The Relative WIS compares the WIS value of each model to the Flusight-baseline model. All models with a relative WIS score less than one outperformed the baseline model when evaluated solely upon WIS. 95% and 50% coverage values are provided for the percent of times that reported weekly incidence values were within the 95% or 50% prediction intervals, respectively, across all the forecast targets submitted by each team. The percent of forecasts submitted is determined by the number of forecast targets submitted by each team out of all possible forecast targets occurring within the duration of the evaluation period. See Supplementary Table 1 for additional model details. [ENS]Ensemble, [STAT]Statistical, [MECH]Mechanistic, [AI/ML]Artificial Intelligence/Machine Learning.

directly inform influenza preparedness and response efforts. During the time period that these forecasting results cover, data were reported daily, with mandatory reporting for influenza admissions from most hospitals in each state, U.S. territories, and D.C starting February 2, 2022. Despite challenges accompanying the shift to the new target of influenza hospitalizations, such as limited historic data from this system for model training, these forecasts provided substantial utility and reinforced a number of lessons learned over the course of previous forecasting activities, both during the pre-pandemic influenza seasons and the COVID-19 pandemic.

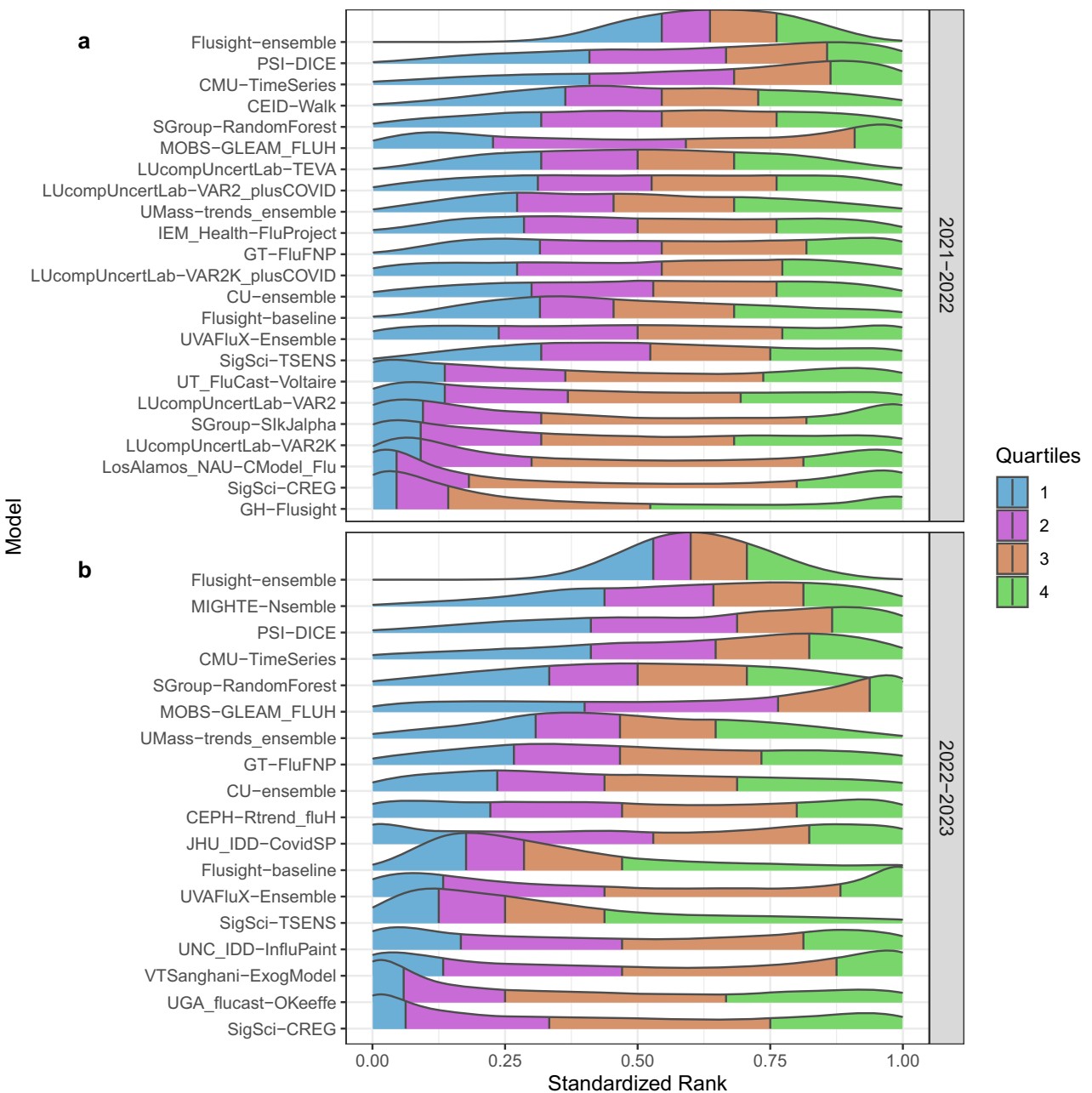

**Fig. 2 | Standardized rank by season.** Standardized rank of weighted interval score (WIS) over all forecast jurisdictions and horizons (1- to 4-week ahead), for the FluSight ensemble and each team submitting at least 75% of the forecast targets (see Table 1 for qualifying teams and season metrics) for the 2021–22 (**a**) and 2022–23 (**b**) seasons.

## Forecast performance–accuracy

As demonstrated in this analysis, collaborative forecasting hub approaches provide opportunities to systematically evaluate performance across multiple modeling strategies and enable the creation of ensemble models. Since a particular model's performance often varies within and across seasons[13], it is helpful to have a unified representation of model inputs that can be used to quickly assess expected upcoming trends. Additionally, this work indicates that ensemble models may also provide more consistently reliable and well-calibrated forecasts across spatial jurisdictions.

Evaluated models cover mechanistic, statistical, ensemble, and AI or machine learning models (see Table 1 and Supplementary Table 1 for additional information). The diversity of model types among the top-performing models was consistent across seasons. In light of this heterogeneity in top-performing model structures and the many

dimensions of differences across forecasting model it has not yet been possible to identify particular characteristics of individual models that are most often associated with high performance. Individual models often vary greatly in their performance within and across seasons (Fig. 1c–e). Across the evaluation period for both seasons and all forecast jurisdictions, the FluSight ensemble was among the top 5 performing models in terms of Absolute WIS and Relative WIS. Additionally, when considering forecast performance by rank (Fig. 2), the FluSight ensemble more accurately predicted weekly influenza hospital admissions than most contributed models with the majority of the FluSight ensemble forecasts falling within the top 50% of submitted forecasts (Table 1, Fig. 2). While the PSI-DICE, CMU-TimeSeries, and MOBS-GLEAM_FLUH models have more forecasts in the top 25%, they exhibit higher spatial heterogeneity than the FluSight ensemble in forecast performance (Fig. 3). The generally high accuracy of the

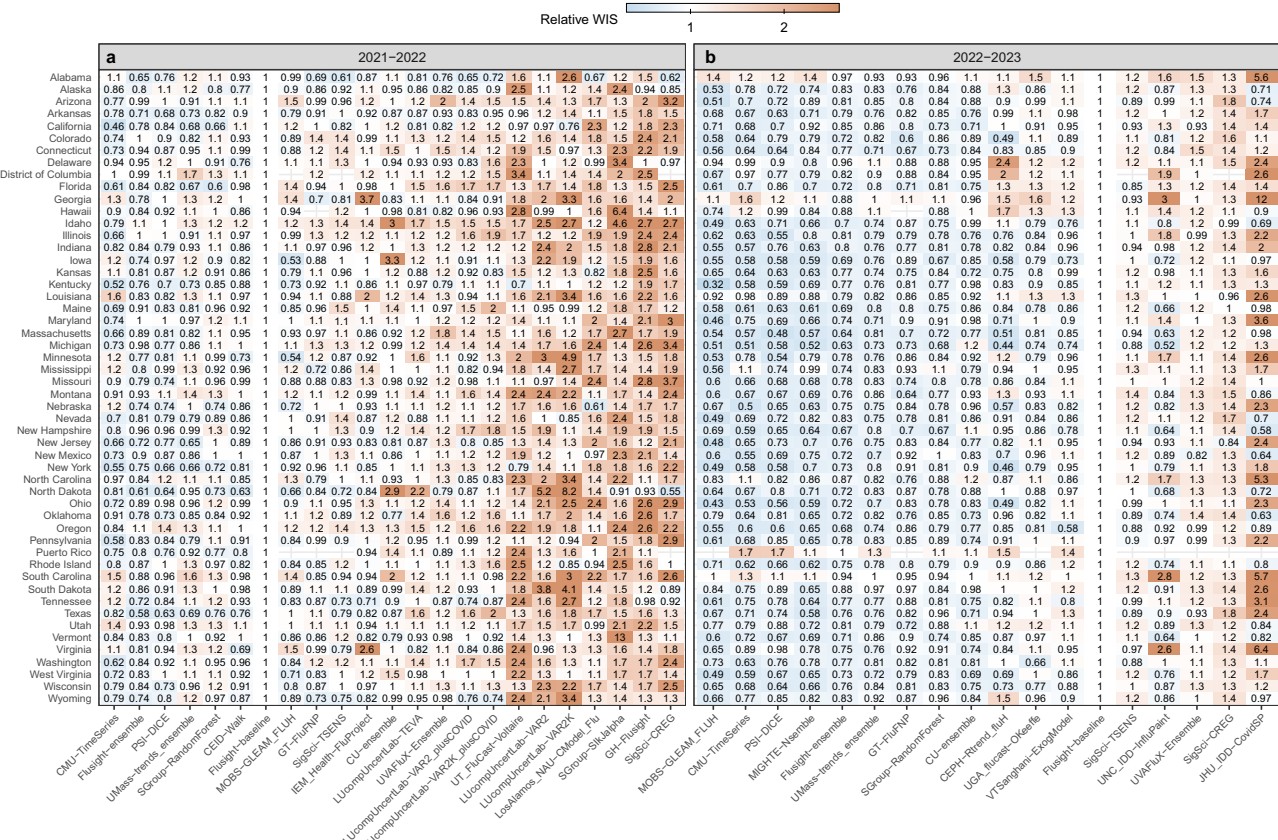

**Fig. 3 | Relative WIS by state and model. State-level WIS values for each team relative to the FluSight baseline model.** The range of Relative WIS values below 1, in blue, indicate better performance than the FluSight baseline (white). Relative WIS values above 1, in red, indicate poor performance relative to the FluSight baseline.

Teams are ordered on horizontal axis from lowest to highest Relative WIS values for each season, 2021–22 (**a**) and 2022–23 (**b**). Analogous jurisdiction-specific relative WIS scores on log transformed counts are displayed in Supplementary Fig. 7.

FluSight Ensemble relative to that of individual models is consistent with previous findings that ensemble models, that utilize the outputs from multiple teams, generally outperform individual models on average[14–17]. Like most models, ensembles may have decreased performance during periods of rapid change when some individual models may have higher accuracy (Fig. 1c, d); however, identifying these time frames and corresponding high-performing models has been difficult a priori[5,6].

One option to better evaluate forecast performance during periods of change and across multiple magnitudes is to evaluate transformed counts[18]. We did not find notable differences in model performance using this approach in either season. We expected that there might be a stronger influence on performance in the 2022–2023 season which saw a sharp increase in hospitalizations in fall 2022, but it is possible that models were not able to capture this initial rise and thus did not accrue additional benefit in the log transform score. The long tail of the season may also have elevated scores across all models.

Forecast model performance tended to decline over longer time horizons. For both the 2021–22 and 2022–23 FluSight seasons, accuracy declined across the 1–4 week ahead horizons. This trend has been observed previously in multiple forecast activities. The U.S. COVID-19 Forecast Hub observed declines in accuracy for forecasted deaths over periods of 1–4 weeks ahead, and German and Polish COVID-19 forecast efforts also showed declines in performance at the 3- and 4-week ahead horizons[19]. Accuracy scores were also shown to decline over longer time horizons for influenza-like-illness forecasts[13].

Across the forecast weeks, individual models often showed larger increases in absolute WIS, while the FluSight ensemble had

the smallest range of absolute WIS for each season, demonstrating one aspect of stability for the FluSight ensemble. In terms of state-level performance, the FluSight ensemble tended to be more robust than individual models, as measured by relative WIS scores (Fig. 3). Similarly, the COVID-19 Forecast Hub ensemble performed better across all locations, with the COVID-19 Hub ensemble being the only model to outperform the baseline in each of the forecast locations[14].

## Forecast performance – coverage

Our analysis found that, as the forecast horizon moved from 1 to 4-weeks, the FluSight ensemble 95% prediction interval coverage declined from 89.61% to 83.74% in 2021–22 and from 85.69% to 77.85% in 2022–23. These results highlight room for improvement in model calibration, as almost all models (with the exception of the UMass trends ensemble) were overconfident in their predictions (Table 2). The lack of comparable historical data for model fitting may have contributed to poor calibration of 95% prediction intervals.

Consistent with past forecasting efforts, forecasting remains difficult in periods of rapid change and epidemic turning points (e.g., during initial increases or periods of peaking activity). This analysis highlights declines in forecast accuracy and coverage during periods of rapid change in influenza hospitalizations during both the 2021–22 and 2022–23 seasons. For example, the only model that had 95% coverage greater than 80% from October to January 2023 when hospitalizations were rapidly increasing and then peaking was LUCompUncertLab-humanjudgment, which did not end up meeting the

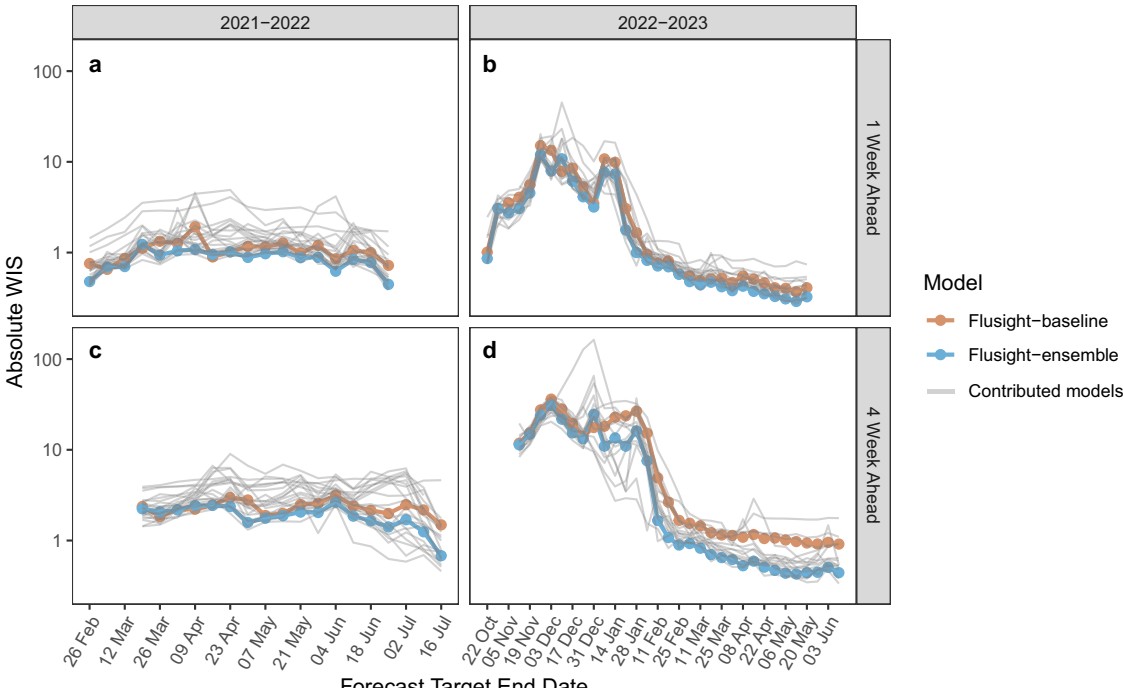

**Fig. 4 | WIS by model.** Time series of log transformed absolute WIS for state and territory targets. Note that the forecast evaluation period translates to 1-week ahead forecast target end dates from February 26–June 25, 2022 (**a**), and October 22, 2022, to May 20, 2023 (**b**), and 4-week ahead forecast target end dates from March 19–July 16, 2022 (**c**), and November 5, 2022–June 10, 2023 (**d**). Weekly results for the FluSight baseline and ensemble models are shown in red and blue respectively. Results for individual contributing models are shown in light gray.

inclusion criteria for the full season analysis. Analogous declines were also observed for COVID-19 case forecasts[20] and mortality forecasts across different waves of the COVID-19 pandemic[14], where forecasts systematically underpredicted during periods of increase and over-predicted during periods of decrease.

Times of changing dynamics are the most important periods for public health response and communication. While forecasting the magnitude at these times may be less tractable, it is possible that we may be able to provide more reliable information during these difficult forecasting periods so that forecasts are better able to inform critical planning. In general, most ensembles tend to predict less activity than observed when trends are steeply increasing and predict more activity than observed when trends are steeply decreasing, especially when there is between- or within-model uncertainty in the timing of peaks in cases, hospitalizations, or deaths. Thus, it may be possible that an ensemble of forecasts for categorical increases or decreases in activity[21] may have additional utility in terms of preserving valuable information while also maintaining the benefits of the use of ensembles over individual models. As such, the FluSight Forecasting Hub added an experimental target in the 2022–23 season for forecasting categorical rate changes in influenza hospitalizations (e.g., probabilities of increase or decrease)[22]. Assessing the utility of this additional forecast target will be an important area of investigation moving forward. Aside from soliciting a separate forecasting target, it may be possible to determine which forecasting models perform better during different phases of epidemics and then use this information to weight models accordingly when their forecasts are aggregated into an ensemble[23].

### Influenza forecasting in the COVID-19 era: challenges and opportunities

Several challenges for forecasting existed during the 2021–22 and 2022–23 influenza seasons. First, as noted earlier, the change in the forecasting target from outpatient ILI percentages to counts of influenza-associated hospitalizations from a data collection system

established during the COVID-19 pandemic meant that there was little data for forecast calibration and training. This shift also required changes in data processing for teams that had produced ILI forecasts previously. While previous data on influenza-associated hospitalizations was available through the FluSurv-NET system, differences in reporting and the spatial resolution, of the FluSurv-NET system may have complicated the process of utilizing this dataset for the purpose of forecasting model calibration. In addition, reporting within the unified HHS-Protect hospitalization dataset changed throughout this forecasting endeavor. For example, the confirmed influenza hospital admissions field only became mandatory for the 2021–22 season on February 2, 2022, leading to an increase in the number of reported hospitalizations and a change in hospital reporting practices during a period of increasing influenza activity.

In addition to changing reporting patterns, the COVID-19 pandemic brought other challenges for forecasting influenza, including changing human behavior. The quantity and types of interactions between people likely changed in tandem with perceptions of the risk of illness with COVID-19. In addition, the use of nonpharmaceutical interventions (NPIs) aimed at preventing SARS-CoV-2 transmission (e.g., stay-at-home orders, mask-wearing) reduced transmission of other respiratory pathogens[9], including influenza. These changes in behavior may be related to the minimal influenza activity observed in the U.S. in the 2020–21 season and the low severity but atypically late influenza season observed in the 2021–22 season. Population-level behavior is difficult to predict, especially in the context of changing public health recommendations and emerging SARS-CoV-2 variants, which complicates the process of forecasting. Despite these challenges, FluSight forecasting teams provided forecasts of confirmed influenza hospitalizations throughout each season, which helped public health officials anticipate trends during the unusually prolonged influenza season in 2021–22, with forecasting efforts extending into June, and then again for the atypically early 2022–23 season.

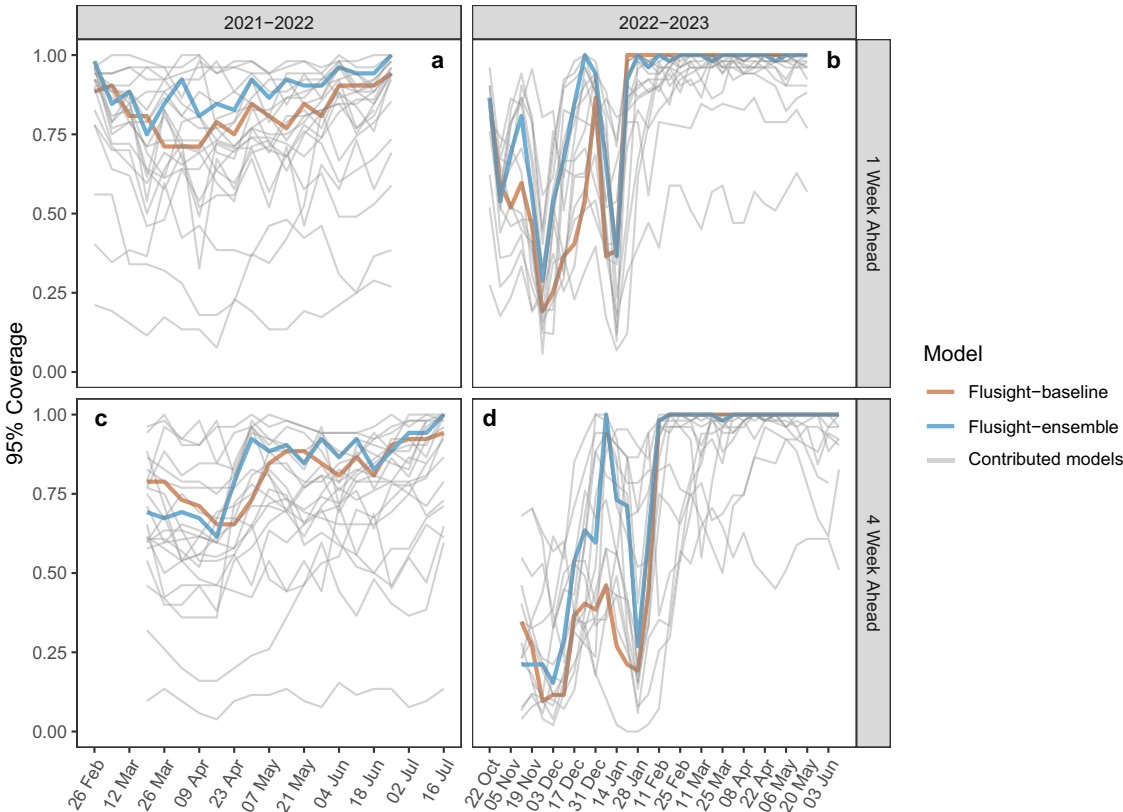

**Fig. 5 | Coverage by model.** 1 and 4-week ahead 95% coverage for state and territory targets. Note that the forecast evaluation period translates to 1-week ahead forecast target end dates from February 26–June 25, 2022 (**a**), and October 22, 2022–May 20, 2023 (**b**), and 4-week ahead forecast target end dates from March 19–July 16, 2022 (**c**), and November 5, 2022–June 10, 2023 (**d**). Weekly results for the FluSight baseline and ensemble models are shown in red and blue, respectively. Results for individual contributing models are shown in light gray.

While the shift to forecasting for a new target presented a modeling challenge, the utility of the corresponding new data source should be recognized[24]. The HHS-Protect dataset[7] provided, in addition to the state-level timeseries, facility-level data, which is at a higher spatial resolution than other indicators of influenza activity. During the forecasting time frame analyzed here, the data were also reported daily with previous-day admission data published as soon as the day after their occurrence, providing a timely source of information. As our data update analysis (Supplementary Figs. 2–4) shows, these data demonstrated remarkably stable reporting behavior, particularly during the 2021–22 season, with 94% of updates resulting in changes of under 10 hospitalizations for subnational jurisdictions. Stability of reporting decreased slightly during the 2022–23 season, with 83% of updates resulting in changes of under 10 hospitalizations for subnational jurisdictions. Degraded forecast performance has been associated with large revisions to initially observed values[6], and consistency in reporting is an important component of a reliable forecasting target. Additionally, this dataset provided national and jurisdictional-level data for confirmed influenza hospital admissions. In contrast with ILI, this indicator eliminated the need to model outpatient visits associated with co-circulating non-influenza pathogens that can cause ILI. The continued availability of rapid, disease-specific indicators of hospitalization, such as those provided by these data, will facilitate improved forecasting utility and possibly improvements in accuracy[25], particularly when forecasts are informed by mechanistic transmission models.

The FluSight forecasting collaboration adapted quickly in 2021 to utilize a novel laboratory-confirmed influenza hospital admission dataset. Even with limited calibration data and atypical influenza seasonality in the 2021–22 and 2022–23 seasons, the FluSight ensemble forecast provided more robust forecasts than individual component models across spatial jurisdictions and time horizons. This result mirrors those of other forecasting hubs. Collaborative hubs also offer the ability for frequent feedback and interaction between modeling teams, providing opportunities for rapidly sharing observations about underlying data and insights for forecast development[26]. We observed poor coverage and general performance, especially at the beginning of the 2022–23 season and during other periods of rapid change. Collective insights from these challenges can also inform when forecasts should be interpreted with extra caution. Ongoing availability of the confirmed influenza hospitalization dataset, which covers all states, could improve model calibration and ultimately contribute to the improvement of influenza forecast performance and utility, as well as continued exploration and improvement of forecasting and ensembling methodologies. These improvements are needed, particularly to more accurately capture trends and appropriate levels of uncertainty during times of rapid change.

## Methods
Forecasts of weekly influenza hospital admissions were openly solicited from existing COVID-19 and influenza forecasting networks every Monday from January 10, 2022, through June 20, 2022, for the 2021–22 season. For the 2022–23 season, forecasts were solicited every Monday from October 17, 2022, through January 9, 2023, and then every Tuesday from January 17, 2023, through May 17, 2023. Weeks were defined in terms of MMWR Epiweeks (EW) spanning Sunday to Saturday[27]. Forecasted jurisdictions included the U.S. national level, all fifty states, Washington D.C., and Puerto Rico. Forecasts for the Virgin Islands, while requested, were not included in this evaluation due to low reported hospitalization counts and irregular

**Table 2 | One-to-four-week coverage and one-to-four-week percent of coverage above 90% for teams meeting inclusion criteria. One-to-four-week is abbreviated with each number and "Wk" indicates week**

| Model | Relative WIS | % WIS Below Baseline | Coverage | | | | % Coverage above 90 | | | |
|---|---|---|---|---|---|---|---|---|---|---|
| | | | 1 Wk | 2 Wk | 3 Wk | 4 Wk | 1 Wk | 2 Wk | 3 Wk | 4 Wk |
| **2021-22** | | | | | | | | | | |
| CMU-TimeSeries | 0.74 | 75.00 | 90.17 | 91.45 | 90.60 | 86.54 | 50.00 | 72.22 | 61.11 | 27.78 |
| Flusight-ensemble | 0.82 | 92.31 | 89.32 | 86.11 | 85.15 | 83.33 | 55.56 | 33.33 | 27.78 | 38.89 |
| PSI-DICE | 0.83 | 76.92 | 88.89 | 83.87 | 78.31 | 76.50 | 38.89 | 27.78 | 5.56 | 0.00 |
| Umass-trends_ensemble | 0.85 | 48.08 | 96.15 | 97.65 | 96.90 | 96.15 | 100.00 | 100.00 | 100.00 | 100.00 |
| Sgroup-RandomForest | 0.91 | 44.23 | 95.41 | 94.87 | 94.66 | 94.12 | 88.89 | 88.89 | 83.33 | 88.89 |
| CEID-Walk | 0.93 | 76.92 | 82.09 | 83.77 | 81.01 | 81.85 | 37.50 | 37.50 | 31.25 | 37.50 |
| Flusight-baseline | 1.00 | 0.00 | 82.26 | 84.19 | 82.48 | 81.62 | 27.78 | 22.22 | 22.22 | 22.22 |
| MOBS-GLEAM_FLUH | 1.02 | 56.00 | 71.11 | 65.80 | 59.79 | 56.49 | 0.00 | 0.00 | 0.00 | 0.00 |
| GT-FluFNP | 1.03 | 54.00 | 70.11 | 68.67 | 68.22 | 70.11 | 5.56 | 16.67 | 16.67 | 22.22 |
| SigSci-TSENS | 1.03 | 46.00 | 74.11 | 73.44 | 70.54 | 69.20 | 11.11 | 5.56 | 5.56 | 5.56 |
| IEM_Health-FluProject | 1.05 | 48.08 | 91.45 | 86.54 | 82.59 | 78.21 | 72.22 | 38.89 | 22.22 | 22.22 |
| CU-ensemble | 1.08 | 32.69 | 79.59 | 80.66 | 76.50 | 71.90 | 16.67 | 11.11 | 0.00 | 0.00 |
| LucompUncertLab-TEVA | 1.20 | 23.08 | 84.86 | 85.58 | 86.06 | 86.18 | 25.00 | 18.75 | 25.00 | 31.25 |
| UVAFluX-Ensemble | 1.27 | 25.00 | 66.05 | 65.51 | 62.58 | 60.95 | 11.11 | 0.00 | 0.00 | 0.00 |
| LucompUncertLab-VAR2_plusCOVID | 1.30 | 38.46 | 76.70 | 74.77 | 73.30 | 70.14 | 17.65 | 5.88 | 5.88 | 5.88 |
| LUcompUncertLab-VAR2K_plusCOVID | 1.39 | 25.00 | 75.72 | 75.24 | 74.04 | 72.72 | 6.25 | 0.00 | 0.00 | 0.00 |
| UT_FluCast-Voltaire | 1.39 | 5.77 | 94.73 | 90.96 | 89.13 | 90.42 | 83.33 | 72.22 | 55.56 | 61.11 |
| LucompUncertLab-VAR2 | 1.53 | 11.54 | 73.87 | 72.29 | 72.17 | 70.81 | 11.76 | 5.88 | 11.76 | 11.76 |
| LucompUncertLab-VAR2K | 1.54 | 11.54 | 81.97 | 81.49 | 83.05 | 85.46 | 6.25 | 18.75 | 25.00 | 37.50 |
| LosAlamos_NAU-Cmodel_Flu | 1.70 | 13.46 | 65.28 | 59.29 | 56.52 | 54.06 | 5.56 | 0.00 | 0.00 | 0.00 |
| Sgroup-SikJalpha | 1.70 | 1.92 | 40.28 | 45.73 | 48.08 | 48.29 | 0.00 | 0.00 | 0.00 | 0.00 |
| GH-Flusight | 1.81 | 5.77 | 18.33 | 12.90 | 11.99 | 10.63 | 0.00 | 0.00 | 0.00 | 0.00 |
| SigSci-CREG | 1.97 | 12.00 | 46.87 | 43.98 | 43.86 | 43.13 | 0.00 | 0.00 | 0.00 | 0.00 |
| **2022-23** | | | | | | | | | | |
| MOBS-GLEAM_FLUH | 0.61 | 94.12 | 81.34 | 77.50 | 76.84 | 78.23 | 41.94 | 29.03 | 29.03 | 25.81 |
| CMU-TimeSeries | 0.67 | 86.54 | 86.27 | 87.12 | 87.25 | 86.73 | 58.06 | 64.52 | 70.97 | 70.97 |
| PSI-DICE | 0.70 | 92.31 | 88.03 | 81.27 | 77.17 | 75.37 | 64.52 | 67.74 | 64.52 | 58.06 |
| MIGHTE-Nsemble | 0.73 | 94.23 | 86.16 | 84.22 | 81.71 | 76.80 | 63.33 | 60.00 | 66.67 | 60.00 |
| Flusight-ensemble | 0.77 | 100.00 | 85.79 | 81.64 | 78.78 | 77.85 | 64.52 | 67.74 | 64.52 | 61.29 |
| Umass-trends_ensemble | 0.80 | 92.31 | 90.88 | 89.89 | 87.41 | 86.17 | 77.42 | 74.19 | 70.97 | 70.97 |
| GT-FluFNP | 0.81 | 92.00 | 75.98 | 72.70 | 75.00 | 77.30 | 55.17 | 55.17 | 55.17 | 65.52 |
| SGroup-RandomForest | 0.82 | 96.15 | 90.06 | 84.49 | 81.86 | 80.38 | 73.33 | 70.00 | 70.00 | 66.67 |
| CU-ensemble | 0.83 | 63.46 | 71.60 | 71.38 | 69.90 | 67.75 | 46.15 | 53.85 | 53.85 | 53.85 |
| CEPH-Rtrend_fluH | 0.84 | 71.15 | 75.82 | 80.22 | 79.33 | 78.02 | 46.43 | 50.00 | 57.14 | 46.43 |
| UGA_flucast-Okeeffe | 0.93 | 66.67 | 80.20 | 73.07 | 68.95 | 66.99 | 50.00 | 46.67 | 40.00 | 40.00 |
| VTSanghani-ExogModel | 0.98 | 34.62 | 65.62 | 61.54 | 58.00 | 58.31 | 0.00 | 0.00 | 0.00 | 4.00 |
| Flusight-baseline | 1.00 | 0.00 | 78.72 | 74.26 | 71.34 | 69.85 | 58.06 | 58.06 | 58.06 | 58.06 |
| SigSci-TSENS | 1.00 | 42.00 | 76.31 | 74.12 | 72.93 | 71.33 | 54.84 | 54.84 | 54.84 | 58.06 |
| UNC_IDD-InfluPaint | 1.05 | 80.39 | 75.12 | 74.18 | 75.12 | 75.82 | 52.00 | 44.00 | 64.00 | 56.00 |
| UVAFluX-Ensemble | 1.11 | 5.88 | 42.88 | 43.53 | 39.35 | 39.80 | 0.00 | 0.00 | 0.00 | 0.00 |
| SigSci-CREG | 1.33 | 14.00 | 68.28 | 62.27 | 58.85 | 56.87 | 48.39 | 48.39 | 45.16 | 45.16 |
| JHU_IDD-CovidSP | 1.88 | 31.37 | 86.74 | 81.67 | 78.18 | 73.60 | 65.38 | 61.54 | 53.85 | 48.00 |

% WIS Below Baseline shows the percent of WIS values for each model below the corresponding Flusight-baseline WIS. The '% Coverage above 90' columns show the percent of weekly 95% coverage values that are greater than or equal to 90% for each model by horizon. Modeling teams are ordered within each season by their relative WIS performance. Season is indicated in bold in the model column.

data submission. Each week, forecasting teams were asked to provide jurisdiction-specific point estimates and probabilistic predictions for 1, 2, 3, and 4-week ahead weekly counts of confirmed influenza hospital admissions. A total of 23 quantiles were requested for the probabilistic forecasts: 0.010, 0.025, 0.050, 0.100, 0.150, ..., 0.950, 0.975, and 0.990. Teams were not required to submit forecasts for all four weeks

ahead or for all locations. Additional details of the forecast submission process (e.g., file formatting, submission procedures, and required metadata) are provided in the FluSight-forecast-data GitHub Repository[22].

The FluSight Ensemble model was generated for all forecasted jurisdictions each week using the unweighted median of each quantile

among eligible forecasts. Forecasts were considered eligible for inclusion in the ensemble if they were submitted by 11:59 PM ET on the due date and if all requested quantiles were provided. Modeling teams could further designate whether a particular model's forecasts should be included in the ensemble. If a forecast was designated as "other", it was not included in the FluSight ensemble and not evaluated in this manuscript.

Baseline forecasts and their prediction intervals were generated each week using the 'quantile baseline' method in the simplets R package[28] based on the incident hospitalizations reported in the previous week, with underlying methodology described as follows. The median prediction of the baseline forecasts is the corresponding target value observed in the previous week, and noise around the median prediction is generated using positive and negative 1-week differences (i.e., differences between consecutive reports) for all prior observations, separately for each jurisdiction. Sampling distributions were truncated to prevent negative values. The same median prediction is used for the 1-through 4-week ahead forecasts. The baseline model's prediction intervals are generated from a smoothed version of this distribution of differences[14,29].

For inclusion in this analysis, forecasting teams must have submitted greater than or equal to 75% of the requested targets, for subnational jurisdictions, between the forecast evaluation period of February 21, 2022, to June 20, 2022 (total of 18 weeks) for 2021–22 or October 17, 2022, to May 15, 2023 (total of 30 weeks) for 2022–23. These periods translate to 4-week ahead forecast target end dates of March 19, 2022–July 16, 2022 for the 2021–22 season and November 11, 2022–June 10, 2023 for the 2022–23 season. The start date of the evaluation period for the 2021–22 season was chosen to be the first forecast date following two weeks of mandatory reporting of confirmed influenza hospitalizations[8] to minimize the potential effects of reporting changes on forecasts. For 2021-22 and 2022-23, three and 12 models were excluded from the primary analysis, respectively, for not meeting the inclusion criteria.

Forecasts were evaluated against the reported number of the previous day's laboratory-confirmed influenza admissions (Field #34) from the COVID-19 Reported Patient Impact and Hospital Capacity by State Timeseries[7,8], with data shifted one day earlier to align with admission date and then aggregated to the weekly scale (from Sunday to Saturday)[22], using data as of September 12, 2022, for 2021–22 and June 13, 2023, for 2022–23. This dataset is subject to revision by submitting facilities; therefore, we analyzed backfill and revision for each season (Supplementary Analysis 1). For each of the contributed forecasts included in the analysis, values were rounded to more closely relate the values of prediction intervals of forecasts to the reported numbers of hospital admissions. In particular, forecast values for quantiles less than 0.5 were rounded down, values for quantiles greater than 0.5 were rounded up, and values for the 0.5 quantiles were rounded normally. This rounding procedure ensured that teams were not penalized for missing the prediction interval by less than one hospital admission.

To evaluate forecast performance across all states, D.C., and Puerto Rico, we primarily used the Weighted Interval Score (WIS). The WIS is a proper score that generates interval scores for probabilistic forecasts provided in the quantile format[14,19]. Briefly, interval scores are used to account for dispersion, underprediction, and overprediction. Forecasts with lower absolute WIS values are considered more accurate than forecasts with higher absolute WIS values. The relative WIS computes the ratio of average WIS values for each pair of models on the subset of forecasts that both models provided and then normalizes by the mean pairwise WIS ratio for the baseline model (See "Supplementary Methods"). Relative WIS values were calculated using the scoringutils package[30]. Simple means were calculated for absolute and relative WIS to get a score for each model, location, and season. Median absolute error (MAE) values are also considered for characterizing differences between forecasted and reported weekly hospitalizations[14]. Unless

otherwise specified, forecasts of national hospitalizations were not included in summary metrics for accuracy (e.g., absolute WIS) since these forecasts can have a disproportionate impact on the overall score. To address concerns related to assessing measures of absolute error on a natural scale when forecasts span multiple orders of magnitude[18], we performed an analogous analysis on log-transformed hospitalization counts after adding one to all counts to account for zero counts (Supplementary Analysis 2). We also performed a separate analysis including only national forecasts (Supplementary Analysis 3).

In addition, we considered coverage values of the quantile-based prediction intervals to assess each model's ability to appropriately capture uncertainty in forecasts. Coverage values are defined as the percent of observed values that fall within the 50% or 95% prediction intervals for the corresponding date. Ideally, the percent coverage values will be equal to the corresponding prediction interval, e.g., 95% percent prediction intervals should contain the reported value 95% of the time.

Comparing model forecasts is complicated by the fact that not all models submit forecasts for each of the forecast targets and for each forecast week in the evaluation period. To partially account for this, we consider the percentage of forecasts submitted as an indicator of how often and how many different types of forecasts were submitted by each team. Following Cramer et al.[14], we also consider a standardized rank score that uses the number of models forecasting a particular location and target and then ranks these forecasts. Ranks were determined by relative WIS performance, with the best-performing model for each observation being assigned a rank of 1 and the worst-performing model receiving a rank equal to the number of models submitting a forecast for the observation. These ranks were standardized by rescaling so that 0 corresponds to the worst rank and 1 corresponds to the best rank.

All analyses were conducted using the R language for statistical computing (version 4.0.3)[31] with scoringutils (version 1.2.2) to generate scores[30].

### Reporting summary

Further information on research design is available in the Nature Portfolio Reporting Summary linked to this article.

## Data availability

The forecast data for each model are available from the FluSight Forecast Hub GitHub repository (https://github.com/cdcepi/Flusight-forecast-data; https://doi.org/10.5281/zenodo.12686773)[22] and the Zoltar forecast archive[12] (https://zoltardata.com/project/299 /viz). These are both publicly accessible. The target data are also available as daily counts for each jurisdiction from HHS[7].

## Code availability

The code used to generate all figures and tables in the manuscript are available in a public repository (https://github.com/cdcepi/FluSight-manuscripts, https://doi.org/10.5281/zenodo.12625724)[32].

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

## Acknowledgements

The authors would like to acknowledge Michael A. Johansson and Nicole Samay for their contributions to this work. M.B.N., P.R., J.T., S.V., A.A., G.K., B.H., B.L.L., M.V.M., M.A.A, A.Srivastava disclose support for the research of this work from the Centers for Disease Control and Prevention (CDC) and Council of State and Territorial Epidemiologists (CSTE), [Cooperative Agreement number NU380T000297] M.C., J.D, K.M., X.X, A.P.P., AV, P.C.V., A.G.K. M.L., M.A. disclose support for the research of this work from the HHS/CDC 6U01IP001137 and HHS/CDC 5U01IP0001137. B.A.P., A.R., H.P.K., Z.Z., G.G., P.A., S.S.B., R.Raman, disclose support for the research of this work from NSF (Expeditions CCF-1918770, CAREER IIS-2028586, RAPID IIS-2027862, Medium IIS-1955883, Medium IIS-2106961, PIPP CCF-2200269), CDC MInD program, faculty gifts from Facebook/Meta, and funds/computing resources from Georgia Tech and GTRI. M.Santillana, L.C. F.L and A.G.M. disclose support for the research of this work from the Centers for Disease Control and Prevention (CDC) and Council of State and Territorial Epidemiologists (CSTE), [Cooperative Agreement number NU380T000297] M.Santillana discloses support for the research of this work from the National Institutes of Health (grant number R01GM130668) and (in part) by contract 200-2016-91779 and cooperative agreement CDC-RFA-FT-23-0069 with the Centers for Disease Control and Prevention. S.V., A.A., G.K., B.H., B.L.L., M.V.M., also disclose support for the research of this work from NSF Expeditions CCF-1918656, VDH Grant VDH-21-501-0135, University of Virginia Strategic Investment Fund Award SIF160. L.C.B., A.Green, A.J.H., D.J.M., R.Rosenfeld, D.S., R.J.T. disclose support for the research of this work from the Centers for Disease Control and Prevention U011P001121 and Centers for Disease Control and Prevention 75D30123C15907. B.T.S., S.A.S., H.L.G., and P.Baccam disclose support for the research of this work from the Centers for Disease Control and Prevention (CDC) and Council of State and Territorial Epidemiologists (CSTE), [Cooperative Agreement number NU38OT000297] N.R. discloses support from National Science Foundation grants CCF-1918770, NRT DGE-1545362, and OAC-1835660. S.T., C.P.S., A.H. disclose support for the research of this work from the National Science Foundation [2127976]. S.T., C.P.S., A.H., J.Lessler, J.C.L., S.L.L., C.D.M., K.S., S-m.J. disclose support from the Centers for Disease Control and Prevention [200–2016]. J.Lessler and J.C.L. disclose support from the National Institutes of Health (NIH 5R01AI102939). A.Mallela, Y.T.L., and W.S.H. disclose support for the research of this work from Laboratory Directed Research and Development Program at Los Alamos National Laboratory [20220268ER]. W.S.H., R. G. P., S. L., Y. C. disclose support for the research of this work from the National Institute of Health [R01GM111510]. Any use of trade, firm, or product names is for descriptive purposes only and does not imply endorsement by the U.S. Government. The findings and conclusions in this report are those of the authors and do not necessarily represent the views of the Centers for Disease Control and Prevention or the National Institutes of Health.

## Author contributions

S.M.M., A.E.W, M.B, and R.K.B. contributed to conceptualization. S.M.M. and R.K.B. wrote the original draft of the manuscript. S.M.M. and A.E.W.

performed the formal analysis. M.B. and C.R. performed supervision and project administration. All authors contributed modeling data and T.M.L., E.L.M., M.Sun, L.A.W., L.C.B., A.Green, A.J.H., D.J.M., R.Rosenfeld, D.S., R.J.T., S.K., S.P., J.S., R.Y., T.K.Y., P.A., S.B., G.G., H.K., B.A.P., R.Raman, A.R., Z.Z., A.Meiyappan, S.O., P.Baccam, H.L.G., S.A.S., B.T.S., M.A., A.G.K., M.L., P.C.V., S.W., J.N., E.C., A.L.H., S.J., J.C.L., J.Lessler S.L.L., C.D.M., K.S., C.S., S.T., T.M., W.Y., N.B., W.S.H., Y.T.L., A.Mallela, Y.C., S.M.L., J.Lee, R.G.P., A.C.P., C.V., L.C., F.L., A.G.M., M.Santillana, M.C., J.T.D., K.M., A.P.P., A.V., X.X., M.B.N., P.R., J.T., C.H.L., S.J., V.P.N., S.D.T., D.W., A.B., J.M.D., S.J.F., G.C.G., E.S., E.W.T., M.G.C., E.Y.C., A.Gerding, A.Stark, E.L.R., N.G.R., L.S., N.W., Y.W., M.W.Z., M.A.A., A.Srivastava, L.A.M., A.A., B.H., G.K., B.L.L., M.M., S.V., P.Butler, A.F., N.M., and N.R. submitted forecast data for the analysis. All authors contributed to the review and editing of the manuscript.

## Competing interests

E.W.T. is an employee of Sanofi, which manufactures influenza vaccines. J.S. and Columbia University disclose partial ownership of SK Analytics. J.S. discloses consulting for BNI. The remaining authors declare no competing interests.

## Additional information

Sarabeth M. Mathis [1,36] ✉, Alexander E. Webber[1,36], Tomás M. León[2], Erin L. Murray[2], Monica Sun[2], Lauren A. White [2], Logan C. Brooks[3,4], Alden Green[3], Addison J. Hu[3], Roni Rosenfeld [3], Dmitry Shemetov[3], Ryan J. Tibshirani[3,4], Daniel J. McDonald [5], Sasikiran Kandula[6], Sen Pei [7], Rami Yaari[7], Teresa K. Yamana [7], Jeffrey Shaman[7,8], Pulak Agarwal[9], Srikar Balusu[9], Gautham Gururajan[9], Harshavardhan Kamarthi[9], B. Aditya Prakash [9], Rishi Raman[9], Zhiyuan Zhao[9], Alexander Rodríguez [10], Akilan Meiyappan[11], Shalina Omar[11], Prasith Baccam[12], Heidi L. Gurung[12], Brad T. Suchoski[12], Steve A. Stage[13], Marco Ajelli[14], Allisandra G. Kummer[14], Maria Litvinova[14], Paulo C. Ventura[14], Spencer Wadsworth[15], Jarad Niemi [15], Erica Carcelen[16], Alison L. Hill [16], Sara L. Loo [16], Clifton D. McKee [16], Koji Sato [16], Claire Smith[16], Shaun Truelove [16], Sung-mok Jung [17], Joseph C. Lemaitre [17], Justin Lessler [17], Thomas McAndrew [18], Wenxuan Ye[18], Nikos Bosse [19], William S. Hlavacek [20], Yen Ting Lin [20], Abhishek Mallela [20], Graham C. Gibson[20], Ye Chen[21], Shelby M. Lamm[21], Jaechoul Lee [21], Richard G. Posner[21], Amanda C. Perofsky [22], Cécile Viboud [22], Leonardo Clemente[23], Fred Lu [23], Austin G. Meyer[23], Mauricio Santillana [23], Matteo Chinazzi [23], Jessica T. Davis[23], Kunpeng Mu[23], Ana Pastore y Piontti[23], Alessandro Vespignani[23], Xinyue Xiong[23], Michal Ben-Nun [24], Pete Riley[24], James Turtle [24], Chis Hulme-Lowe[25], Shakeel Jessa [25], V. P. Nagraj[26], Stephen D. Turner [26], Desiree Williams [26], Avranil Basu[27], John M. Drake [27], Spencer J. Fox [27], Ehsan Suez[27], Monica G. Cojocaru[28], Edward W. Thommes[28,29], Estee Y. Cramer [30], Aaron Gerding[30], Ariane Stark[30], Evan L. Ray[30], Nicholas G. Reich[30], Li Shandross [30], Nutcha Wattanachit[30], Yijin Wang [30], Martha W. Zorn[30], Majd Al Aawar[31], Ajitesh Srivastava [31], Lauren A. Meyers [32], Aniruddha Adiga[33], Benjamin Hurt [33], Gursharn Kaur[33], Bryan L. Lewis [33], Madhav Marathe[33], Srinivasan Venkatramanan [33], Patrick Butler [34], Andrew Farabow[34], Naren Ramakrishnan[34], Nikhil Muralidhar[35], Carrie Reed[1], Matthew Biggerstaff [1] & Rebecca K. Borchering [1] ✉

[1]Centers for Disease Control and Prevention, Atlanta, GA, USA. [2]California Department of Public Health, Richmond, CA, USA. [3]Carnegie Mellon University, Pittsburgh, PA, USA. [4]University of California, Berkeley, Berkeley, CA, USA. [5]University of British Columbia, Vancouver, BC, Canada. [6]Norwegian Institute of Public Health, Oslo, Norway. [7]Columbia University, New York, NY, USA. [8]Columbia University School of Climate, New York, NY, USA. [9]Georgia Institute of Technology, Atlanta, GA, USA. [10]University of Michigan, Ann Arbor, MI, USA. [11]Guidehouse Advisory and Consulting Services, McClean, VA, USA. [12]IEM, Bel Air, MD, USA. [13]IEM, Baton Rouge, LA, USA. [14]Indiana University School of Public Health, Bloomington, IN, USA. [15]Iowa State University, Ames, IA, USA. [16]Johns Hopkins University, Baltimore, MD, USA. [17]University of North Carolina at Chapel Hill, Chapel Hill, NC, USA. [18]Lehigh University, Bethlehem, PA, USA. [19]London School of Health and Tropical Medicine, London, UK. [20]Los Alamos National Laboratory, Los Alamos, NM, USA. [21]Northern Arizona University, Flagstaff, AZ, USA. [22]Fogarty International Center, National Institutes of Health, Bethesda, MD, USA. [23]Northeastern University, Boston, MA, USA. [24]Predictive Science Inc, San Diego, CA, USA. [25]Signature Science, LLC, Austin, TX, USA. [26]Signature Science, LLC, Charlottesville, VA, USA. [27]University of Georgia, Athens,

GA, USA. [28]University of Guelph, Guelph, ON, Canada. [29]Sanofi, Toronto, ON, USA. [30]University of Massachusetts Amherst, Amherst, MA, USA. [31]University of Southern California, Los Angeles, CA, USA. [32]University of Texas Austin, Austin, TX, USA. [33]University of Virginia, Charlottesville, VA, USA. [34]Virginia Tech, Arlington, VA, USA. [35]Stevens Institute of Technology, Hoboken, NJ, USA. [36]These authors contributed equally: Sarabeth M. Mathis, Alexander E. Webber. ✉e-mail: nqr2@cdc.gov; xhq2@cdc.gov

