## [Peer Review File · Nature Communications]

Evaluation of FluSight influenza forecasting in the 2021-22 and 2022-23 seasons with a new target laboratory-confirmed influenza hospitalizationsREVIEWER COMMENTS

Reviewer #1 (Remarks to the Author):

The paper provides a well-executed evaluation of the novel version of the Flusight competitions during the last two seasons in the US. The Flusight competitions have been the central CDC-coordinated effort for collaborative real-time short-term forecasting of influenza. The novelty in the competition design arises from a change to a new target as a result of the Covid pandemic moving from weakly reported symptoms (ILI percentages) to confirmed weakly test cases (absolute number).

Major Remarks:

1. The paper is very much focused on score-based evaluation. This is of course important, but I think it is equally important to give an impression of what forecasts actually looked like. Figure 1 does this to some degree, but only for one model, the national level and four different time points. I suggest including some more displays of actual forecasts. For instance, the following aspects could be shown:

- diversity/heterogeneity of forecasts from different models at some interesting points in time covering changes in reporting system of the confirmed test cases and relation to NPIs
- behavior at the season onset
- illustration of spatial heterogeneities in performance
- displays of forecasts at fixed horizons (rather than with fixed forecast origins as in Fig 1). These plots would show an overlay of the actual time series and the forecasts for the respective weeks as issued 1, 2, 3 or 4 weeks before. An example of such a plot can be found in Figure 5 of Ray et al (2016) <https://doi.org/10.1002/sim.7488>

I think some more such figures could give the reader a better impression of what the Hub actually produced and could provide guidance for public health officials of whether they would consider the outputs helpful.

2. It would be interesting to see how the considered target of confirmed cases related to the recorded ILI-values, in particular for judging reporting delays by the new system more effectively. From the backfill analysis in the supplementary material it seems that the change in reporting system in February 2022 yielded to some extended more precise initial reported values and quicker updates to the final values. But the strange pattern of over- rather than underreporting in the initial values persists (Why?). How does the evaluation of the forecast study account for these effects given the different quality of last available observations in the two reporting regimes? How does this translate to the subnational analysis? Figures S3 and S4 suggest that there were dates of substantial reporting outliers. It would be of interest to study forecast behavior around and exempt those events and in detail in particular when before the reporting regime change in February 2022. As teams update models according to latest previous performance, what would be the suggested delay for a somewhat stable real-time evaluation in future rounds of the competition?

Minor points:

1. I am somewhat confused by the use of the relative WIS. In the paper by Cramér et al (2021, PNAS) this quantity was introduced to compare models to a baseline model, but also to account for partial missingness of forecasts. It is obtained by taking ratios of mean WIS values in a specific fashion (compare the supplementary material of Cramér et al). It is thus not a score that can be evaluated for individual pairs of forecasts and observations. In line 182, however, it is stated that means of relative WIS values were taken, which is not coherent with the previous use of relative WIS. Could the authors state explicitly how they computed the relative WIS and what sort of averaging was performed? This can be moved to a Supplement, but it should be documented explicitly. In the main manuscript, some intuition of what relative WIS computes should be added. In the original version this would be something like "the ratio of mean WIS of a given model and a baseline model, with a correction for the fact that not all models covered the same set of forecast targets. Values below one indicate better performance than the baseline model, values above one worse performance".

2. Relatedly to the above, the original way of using the "relative WIS" was intended to account for

the difficulty stated in lines 200 and following ("Comparing model forecasts is complicated by the fact that not all models submit forecasts for each of the forecast targets and for each forecast week in the evaluation period.") Could the authors clarify how their use of relative WIS relates to this problem?

3. Line 246: A relative WIS of 12 is quite extraordinary. Can the authors briefly comment on what led any model to be outperformed by a simple baseline model by such a staggering margin?

4. Line 290: I am not sure the comparison of absolute WIS values across different time points is so meaningful as the WIS scales with the order of magnitude of the quantity to be predicted. So it is largely due to the way WIS is defined that high values occur when influenza numbers are high and low values occur when influenza numbers are low.

5. I could not find any reference to a preregistration of the conducted real-time study. That would have certainly added value to the provided analysis.

6. The experimental categorical target seems interesting. Unfortunately, I could not find any evaluation on that.

Reviewer #2 (Remarks to the Author):

This paper provides a detailed evaluation of the performance of FluSight influenza forecasting models for hospitalizations during the 2021-2023 period, with a particular focus on the FluSight ensemble model in comparison to other component models. The manuscript is overall well-written. However, several areas could be enhanced to improve clarity and utility for the reader. My comments are as follows:

1. The paper offers a comprehensive analysis, yet it lacks a clear guide for readers on the optimal model selection for specific scenarios. A more systematic stratification of the models could significantly aid in understanding their applicability. For instance, Table 1 could be organized to distinguish between model types, such as statistical/machine learning, mechanistic, and ensemble models. This organization would allow readers to more easily identify the strengths and weaknesses of each model type in different forecasting contexts.

2. The focus on the FluSight ensemble model's performance is evident, particularly in Figure 1. However, this emphasis may overshadow the achievements of other models that also performed well. For example, the abstract mentions, "Averaging across all forecast targets, the FluSight ensemble was the 2nd most accurate model measured by WIS in the 2021-22 and the 5th most accurate in the 2022-23 season." Expanding the discussion to include these other top-performing models would provide a more balanced view of the available forecasting tools and their relative efficacies across different seasons.

3. Related to the first point, some insights/discussion into why certain models outperform others could greatly benefit practitioners in selecting the most appropriate model for specific circumstances. Understanding the underlying factors contributing to a model's success or limitations, such as data inputs or model structure, would be useful.

Reviewer #3 (Remarks to the Author):

SUMMARY

The manuscript describes the performance of the FluSight influenza forecasting hub and the corresponding FluSight ensemble model over the two influenza seasons 2021/22 and 2022/23. The results emphasize the quality and reliability of the ensemble forecast via different outcome metrics, especially in comparison with a predefined baseline model.

GENERAL COMMENTS

While the results and methodology used in the FluCast Ensemble are highly relevant to a range of readers, from epidemiologists to public health policy makers, the manuscript currently lacks a clearly defined research question, in my view. This also makes it rather difficult to assess the innovative content of the manuscript: the general methodology (ensemble forecast, outcome metrics, ...) is not new and has already been published umpteen times, especially by the many COVID-19 Hubs. Even the FluCast Hub itself and its results have been published several times (reference [20] is conceptually very similar to the submitted manuscript). Accordingly, I would welcome it if the research question, the research need, the innovative content and the overall research findings were more clearly defined in order to strengthen the text, which in my opinion currently has the format of an "annual report" rather than a research paper.

As with many other forecasting or scenario hubs, my biggest concern regarding the proposed method(s) is the understanding and the correctness of the ground truth, i.e. the underlying hospitalization data. Hereby, I would be very interested if the hub understands and attempts to clean bias from the collected data to avoid garbage-in-garbage-out forecasts. Having a quick look at reference [8] I could imagine a variety of different biases which might influence the direct causality between illnesses and hospitalisations. To name a few: regionally varying willingness to report (good quality) data, different reporting delays and changing reporting patterns, socially/regionally biased patient selection via CMS hospitals, inaccurate differences between primary and secondary diagnosis (how do they count a bypass patient tested positive for influenza?), or limited bed capacities (if all beds are occupied, the counts will not increase anymore, independent of the disease trend).

In any case it would increase confidence in the results - in particular if they are said to be politically relevant - if a section was added in which the data collection process and in particular the involved weaknesses is described.

SPECIFIC COMMENTS

137) Please elaborate this in more detail. Who could vote to in/exclude a certain model's forecast and how?

141-142) A "package" cannot generate a forecast. Please state the used method (from the package).

151) Please, state how the ensemble members were casted. E.g. were they paid, if not, what was their motivation to contribute? Did they have the same system knowledge (e.g. modeling team from CDC might have better insights than a hackathon-team from a countryside highschool)?

248) FluSight hub ensemble = FluSight Ensemble? In general, since both FluSight models, the baseline and the ensemble, start with the same prefix (namely FluSight), it is sometimes a little difficult to differentiate between the two when reading. However, since they have fundamentally different roles, it would be good to be able to distinguish between the two quickly and clearly. I would suggest introducing appropriate abbreviations (e.g. BaseFlu, EnsembleFlu) This would make the result section much easier to read.

387) I guess you want to refer to Figure 4 instead?

Tables 1 and 2) Having some background information on the used methods in the specific models would drastically increase the informative value of these tables. Currently, it is only some ranked models of which we know nothing about.

Figure 3) First, in my opinion, this should be a table. Second, there is a lot of content in this table which is not relevant for the manuscript main text. Since the discussion only refers to the spatial heterogeneity, maybe you find a different way to depict this visually. Finally, the font size is too small, but this is rather a cosmetic comment and you probably know that already.

References) the link to [8] is broken in the meanwhile

Response to REVIEWER COMMENTS

Reviewer #1 (Remarks to the Author):

The paper provides a well-executed evaluation of the novel version of the Flusight competitions during the last two seasons in the US. The Flusight competitions have been the central CDC-coordinated effort for collaborative real-time short-term forecasting of influenza. The novelty in the competition design arises from a change to a new target as a result of the Covid pandemic moving from weakly reported symptoms (ILI percentages) to confirmed weakly test cases (absolute number).

Major Remarks:

1. The paper is very much focused on score-based evaluation. This is of course important, but I think it is equally important to give an impression of what forecasts actually looked like. Figure 1 does this to some degree, but only for one model, the national level and four different time points. I suggest including some more displays of actual forecasts. For instance, the following aspects could be shown:

- diversity/heterogeneity of forecasts from different models at some interesting points in time covering changes in reporting system of the confirmed test cases and relation to NPIs
- behavior at the season onset
- illustration of spatial heterogeneities in performance
- displays of forecasts at fixed horizons (rather than with fixed forecast origins as in Fig 1). These plots would show an overlay of the actual time series and the forecasts for the respective weeks as issued 1, 2, 3 or 4 weeks before. An example of such a plot can be found in Figure 5 of Ray et al (2016)

<https://doi.org/10.1002/sim.7488>

I think some more such figures could give the reader a better impression of what the Hub actually produced and could provide guidance for public health officials of whether they would consider the outputs helpful.

We thank the reviewer for this comment and have added more plots of individual forecasts with corresponding observed data to the supplement. We have also added a link to an interactive dashboard where individual forecasts can be viewed with retrospective data current at the time forecasts were made and the final data. (<https://zoltardata.com/project/299/viz>) Due to the relatively large number of submitting teams we think that this interactive dashboard is the best way for readers to interact with individual forecasts.

For the illustration of heterogeneities in performance across spatial jurisdictions, we refer the reviewer to Figure 3. This information is also being provided in table format (see reviewer response below). We have also included in the response PDF time series for all the locations and all the models for forecast weeks ending 11/14/22, 12/5/22, and 2/27/23. The first two of which correspond to periods of rapid changes (increase and a peak) and the last to a period with less rapid change. We defer to the editors whether this would be a helpful addition to the supplement.

We have detailed in the discussion the difficulties of starting forecasting for a new forecast target and hope that this helps provide context for regarding the forecasts with different levels of uncertainty.

Further we describe in the results and discussion periods of time when the forecasts tend to display decreases in performance which we hope informs the decision process for what level of trust to put in forecast outputs at different times of the seasonal epidemics.

2. It would be interesting to see how the considered target of confirmed cases related to the recorded ILI-values, in particular for judging reporting delays by the new system more effectively. From the backfill analysis in the supplementary material it seems that the change in reporting system in February 2022 yielded to some extended more precise initial reported values and quicker updates to the final values. But the strange pattern of over- rather than underreporting in the initial values persists (Why?). How does the evaluation of the forecast study account for these effects given the different quality of last available observations in the two reporting regimes? How does this translate to the subnational analysis? Figures S3 and S4 suggest that there were dates of substantial reporting outliers. It would be of interest to study forecast behavior around and exempt those events and in detail in particular when before the reporting regime change in February 2022. As teams update models according to latest previous performance, what would be the suggested delay for a somewhat stable real-time evaluation in future rounds of the competition?

Given that reporting influenza hospital admissions to HHS Protect was optional prior to February 2022, examining forecast performance using these data would be of limited interest and value. During periods of optional reporting, we have noticed low levels of hospital reporting, making a comparison with forecasted values, which assume complete reporting, unhelpful to compare. We didn't include forecasts from before February 2022 and did include a 2-week buffer from when reporting became mandatory. We also communicated with forecasters that it would be the final data that would be comparing the model forecasts to, not the values reported in real time. It is especially difficult to compare influenza like illness (ILI) to hospitalization data during the period of the study since COVID-19 and influenza share many symptoms that fall in the ILI case definition. Care-seeking behavior also changed during the pandemic, making comparisons to past ILI forecasting efforts difficult. We discuss the choice to shift to the new laboratory confirmed influenza hospital admission target and now provide an additional reference that further elaborates on this choice and the complications of considering ILI data since the beginning of the COVID-19 pandemic.

Since this is a real-time forecast analysis, it is not possible to know when values will be updated substantially, so we prefer not to remove these dates from the analyzed results. We ask that forecasters use their best judgment and expressed that the ultimate comparison data would be data that was finalized after the forecasting challenge. We have also highlighted in the main text of the manuscript by adding "There were infrequent larger updates to reported admissions" to lines 236-237 and 246-247 along with specifying that most updates occurred within two weeks of initial publication which is within the expected range of revision time. We prefer not to delve into further detail on the small subset of points where revisions were made.

Minor points:

1. I am somewhat confused by the use of the relative WIS. In the paper by Cramér et al (2021, PNAS) this quantity was introduced to compare models to a baseline model, but also to account for partial missingness of forecasts. It is obtained by taking ratios of mean WIS values in a specific fashion (compare the supplementary material of Cramér et al). It is thus not a score that can be evaluated for individual pairs of forecasts and observations. In line 182, however, it is stated that means of relative WIS values were taken, which is not coherent with the previous use of relative WIS. Could the authors state explicitly how they computed the relative WIS and what sort of averaging was performed? This can be moved to a Supplement, but it should be documented explicitly. In the main manuscript, some intuition of what relative WIS computes should be added. In the original version this would be something like "the ratio of mean WIS of a given model and a baseline model, with a correction for the fact that not all models covered the same set of forecast targets. Values below one indicate better performance than the baseline model, values above one worse performance".

Thank you for this comment, although it was a minor point, upon review of the references, we have decided to update our workflow to use the R package `scoringutils` which uses the pairwise relative WIS calculation that is found more frequently in the literature, including the Cramer et al. paper. Methodological details have been updated in the manuscript on lines 192-197. We also added the appropriate equation to the supplement. In accordance with this update, we have also opted to use the `scoringutils` functionality for the absolute error metric, which is a median value, rather than the original Mean Absolute Error (MAE) that was included in the initial submission of the manuscript. For context, the pairwise relative WIS values changed by a maximum of 0.04 in both seasons, with the lowest ranked model scoring 1.97 in the 21-22 season, and 1.33 in the 22-23 season. There were also no changes to the rankings of the top 7 models in either season. Given these observations, we are confident that our main findings remain unchanged.

In text changes have been made to reflect the minor changes in values in lines 274-275, 284, 286-287, 300, 303, 308, and 310.

Figures 2, 3, Tables 1, 2, Supplementary Figure 1, Supplementary Tables 3, and 4 have also been updated to reflect these minor changes.

2. Relatedly to the above, the original way of using the "relative WIS" was intended to account for the difficulty stated in lines 200 and following ("Comparing model forecasts is complicated by the fact that not all models submit forecasts for each of the forecast targets and for each forecast week in the evaluation period.") Could the authors clarify how their use of relative WIS relates to this problem?

Please see comment above about updated implementation of relative WIS. We think that changes between our original and updated relative WIS values were minimal because we restricted our primary analysis to models that submitted greater than or equal to 75% of forecasts.

Our original relative WIS calculation scored each model's forecasts directly compared against those of the baseline model. Since the FluSight-baseline model has submissions for all forecast targets this ensures that every forecast will have a reference forecast for calculating the corresponding relative WIS.

We now include this information in Table 2.

3. Line 246: A relative WIS of 12 is quite extraordinary. Can the authors briefly comment on what led any model to be outperformed by a simple baseline model by such a staggering margin?

We've reviewed this particular example for both seasons and found that the large relative WIS values were likely due to poor model calibration. The model with that score predicted almost 2000 admissions at one horizon when the observed admissions were less than 10 in the 2021-22 season; There were a few other poorly calibrated forecasts for the same time frame, but on a less extreme scale. Similarly, in 2022-23, the model that scored a relative WIS of 12 predicted 1000-2000 admissions for a few weeks when actual admissions ranged from 138-244. We would prefer not to highlight this rare occurrence in the manuscript as we think it would put undue attention on a small number of forecasts.

In addition, we have added a link to the Zoltar platform reference [21] so that specific model performance can be viewed.

4. Line 290: I am not sure the comparison of absolute WIS values across different time points is so meaningful as the WIS scales with the order of magnitude of the quantity to be predicted. So it is largely due to the way WIS is defined that high values occur when influenza numbers are high and low values occur when influenza numbers are low.

We agree which is why we emphasize the relative values rather than absolute WIS throughout the majority of the manuscript but prefer to provide multiple metrics for review in our analysis.

5. I could not find any reference to a preregistration of the conducted real-time study. That would have certainly added value to the provided analysis.

For this real-time study, forecasting guidance and instructions were provided on GitHub before the start of the forecasting period. We did not register the study in any other repository. Archived versions of the instructions are available in the repository, and all submitted forecasts are time stamped from the time of submission.

6. The experimental categorical target seems interesting. Unfortunately, I could not find any evaluation on that.

We appreciate this comment. We agree that the experimental categorical target work is interesting, but have decided to keep this work separate from this manuscript as it is ongoing. Further analysis and evaluation of the categorical targets are forthcoming. See the abstract included at the end of this document on related work presented at Epidemics 9 in November 2023.

Reviewer #2 (Remarks to the Author):

This paper provides a detailed evaluation of the performance of FluSight influenza forecasting models for

hospitalizations during the 2021-2023 period, with a particular focus on the FluSight ensemble model in comparison to other component models. The manuscript is overall well-written. However, several areas could be enhanced to improve clarity and utility for the reader. My comments are as follows:

1. The paper offers a comprehensive analysis, yet it lacks a clear guide for readers on the optimal model selection for specific scenarios. A more systematic stratification of the models could significantly aid in understanding their applicability. For instance, Table 1 could be organized to distinguish between model types, such as statistical/machine learning, mechanistic, and ensemble models. This organization would allow readers to more easily identify the strengths and weaknesses of each model type in different forecasting contexts.

We have added the general type of model to each model in Table 1 (additional detail is provided in Supplementary Table 1). We prefer not to order by model type, since we have organized individual models by their performance. Additionally, we did not see a clear relationship between model type and performance and have added the following text to lines 257-261 of the Results “Top performing models in the 2021-22 season included statistical, mechanistic and ensemble models. In 2022-23, top performing models included mechanistic, statistical, ensemble, and one machine learning model. There were also statistical, mechanistic, AI or machine learning, and ensemble models among the poorest performing models across seasons.” We thank the reviewer for this comment and could see this being a good direction for future work, particularly when larger differences are seen between the performance of different types of models.

2. The focus on the FluSight ensemble model’s performance is evident, particularly in Figure 1. However, this emphasis may overshadow the achievements of other models that also performed well. For example, the abstract mentions, “Averaging across all forecast targets, the FluSight ensemble was the 2nd most accurate model measured by WIS in the 2021-22 and the 5th most accurate in the 2022-23 season.” Expanding the discussion to include these other top-performing models would provide a more balanced view of the available forecasting tools and their relative efficacies across different seasons.

We thank the reviewer for this comment. As mentioned above, we have added notes on which types of models performed well as well as added the basic model type to Table 1. We also now highlight the top performing models by visualizing their performance across different horizons in the included `FluSight_2021-22_2022-2023_Additional_Reviewer_documents.pdf`. Each model is the intellectual property of the submitting team and we encourage teams to publish about their forecasting efforts and do not want to overstep by publishing details about models from participating groups. We hope that the following link to Zoltar which includes visualizations of model-specific forecasts, <https://zoltardata.com/project/299/viz>, assists with further interrogation of specific model results. Teams were given the opportunity to provide additional information in Supplementary Table 1, including each teams’ repository URL which will contain more detailed model information and code. We have also provided, in the response to reviewers, plots of forecasts for all contributing models and defer to the editors whether it would be helpful to have all or some number of models included in the supplement.

3. Related to the first point, some insights/discussion into why certain models outperform others could greatly benefit practitioners in selecting the most appropriate model for specific circumstances.

Understanding the underlying factors contributing to a model's success or limitations, such as data inputs or model structure, would be useful.

This is a common question among multi-modeling efforts and unfortunately there is not an easy answer. It is difficult to evaluate which components of models contribute to better or worse performance for multiple reasons. For one, it is difficult to capture the complex nature of all of the modeling methods and their characteristics can vary or align in several different dimensions and may also change during the season with finetuning of parameterization or approach. We have also found that it is often a mix of compartmental/mechanistic and statistical approaches that are in the top 5 performing models within and across seasons. We've added the text below to the manuscript to highlight the heterogeneity in model structure observed across top performing models. We've also added a legend to Table 1 that indicates which broad category of model each model falls in, so that readers can see this heterogeneity beyond just the top ranked models.

For similar reasons, it has not yet been possible to characterize what types of models will perform better at different times and under different circumstances as this is highly variable, and epidemic season dependent. That is one reason why we have highlighted the robustness of the ensemble. Since ensemble models have shown to have generally high performance they can be seen as a good option when there is limited to no information about how individual models may perform for a particular epidemic prediction task a priori.

We've added the following text to the accuracy section of the discussion:

"These models cover mechanistic, statistical, ensemble, and AI or machine learning models (see Supplementary Table 1). The diversity of model types among the top performing models was consistent across seasons. In light of this heterogeneity in top performing models structures and the many dimensions of differences across forecasting models it has not yet been possible to identify particular characteristics of individual models that are most often associated with high performance. Individual models often vary greatly in their performance within and across seasons."

Reviewer #3 (Remarks to the Author):

SUMMARY

The manuscript describes the performance of the FluSight influenza forecasting hub and the corresponding FluSight ensemble model over the two influenza seasons 2021/22 and 2022/23. The results emphasize the quality and reliability of the ensemble forecast via different outcome metrics, especially in comparison with a predefined baseline model.

GENERAL COMMENTS

While the results and methodology used in the FluCast Ensemble are highly relevant to a range of readers, from epidemiologists to public health policy makers, the manuscript currently lacks a clearly defined research question, in my view. This also makes it rather difficult to assess the innovative content of the manuscript: the general methodology (ensemble forecast, outcome metrics, ...) is not new and has already been published umpteen times, especially by the many COVID-19 Hubs. Even the FluCast

Hub itself and its results have been published several times (reference [20] is conceptually very similar to the submitted manuscript). Accordingly, I would welcome it if the research question, the research need, the innovative content and the overall research findings were more clearly defined in order to strengthen the text, which in my opinion currently has the format of an “annual report” rather than a research paper.

We thank the reviewer for their comments and observations. We have added the following text to the introduction to formalize the underlying research question, “Our objective is to address the performance of these forecasts in post-COVID influenza seasons which are expected to have atypical timing and intensity. By evaluating forecast performance, particularly considering a new forecast format and dataset with limited calibration data, we can identify specific areas for forecast improvement.”

We have also included the “**Influenza forecasting in the COVID-19 era: challenges and opportunities**” section of the discussion to describe further novelty of this analysis including the context of forecasting when there is atypical influenza seasonality that followed the pause in influenza activity during the COVID-19 pandemic. While FluSight has a track record of soliciting uniform forecasts for almost a decade, there is a general lack of standardization, and likely repeatability, which remain a major challenge in infectious disease forecasting. As others have noted (Pollett et al., 2020, 2021) lack of standard methodology, reporting, and evaluation does make universal cross comparisons or summaries difficult.

Pollett, S., et al. (2021). Recommended reporting items for epidemic forecasting and prediction research: The EPIFORGE 2020 guidelines. *PLoS Medicine*, 18(10), e1003793.
<https://doi.org/10.1371/journal.pmed.1003793>

Pollett, S., et al. (2020). Identification and evaluation of epidemic prediction and forecasting reporting guidelines: A systematic review and a call for action. *Epidemics*, 33, 100400.
<https://doi.org/10.1016/j.epidem.2020.100400>

As with many other forecasting or scenario hubs, my biggest concern regarding the proposed method(s) is the understanding and the correctness of the ground truth, i.e. the underlying hospitalization data. Hereby, I would be very interested if the hub understands and attempts to clean bias from the collected data to avoid garbage-in-garbage-out forecasts. Having a quick look at reference [8] I could imagine a variety of different biases which might influence the direct causality between illnesses and hospitalisations. To name a few: regionally varying willingness to report (good quality) data, different reporting delays and changing reporting patterns, socially/regionally biased patient selection via CMS hospitals, inaccurate differences between primary and secondary diagnosis (how do they count a bypass patient tested positive for influenza?), or limited bed capacities (if all beds are occupied, the counts will not increase anymore, independent of the disease trend).

We thank the reviewer for their comment. While we recognize the variety of biases that could exist in this dataset, we believe that we picked an actionable target from a dataset that meets the needs described above. This dataset is also used as a main metric for hospital trends for both COVID-19 and influenza at the following websites:

- <https://www.cdc.gov/respiratory-viruses/data-research/dashboard/illness-severity.html>

- https://covid.cdc.gov/covid-data-tracker/#maps_positivity-week
- <https://www.cdc.gov/flu/weekly/index.htm>

As modelers, we always want the cleanest and most accurate data, but with public health data, especially if there is the potential to include protected health information, it can be difficult to acquire appropriate data for the entire United States while still at a usable scale to analyze trends in smaller jurisdictions. This dataset is also shared publicly in a manner that stores revisions to the data, allowing modelers outside of the government to participate in forecasting efforts and understand how reporting patterns changed over two influenza seasons varying timing and intensity.

We've also added the following reference to a recently accepted commentary that includes additional detail on the hospital admission dataset:

Borchering R.K., et al. Responding to the return of influenza in the United States: applying CDC surveillance, analysis, and modeling to inform understanding of seasonal influenza. *JMIR Public Health and Surveillance*. (2024). <https://publichealth.jmir.org/2024/1/e54340>.

We looked at available bed capacity data and found that during the timeframe of investigation for this manuscript, there were only a few days in two states where over capacity was reported. In general, values of 50-80% capacity were reported across jurisdictions, so we do not believe this issue would have a substantive impact on these results.

In any case it would increase confidence in the results – in particular if they are said to be politically relevant – if a section was added in which the data collection process and in particular the involved weaknesses is described.

Borchering et al. (2024) mentioned in response to the previous comment above goes into additional detail about why we switched the forecasting target from influenza-like-illness to the novel target of laboratory confirmed influenza hospital admissions. We've added this reference to the manuscript as well.

Given the scope of this manuscript and the widespread use of this dataset, we prefer not to include more details than adding the above reference. We also refer the reviewer to the FAQ document about the dataset referenced in the manuscript and included below.

COVID-19 Guidance for Hospital Reporting and FAQs For Hospitals, Hospital Laboratory, and Acute Care Facility Data Reporting, 11 June 2023, www.hhs.gov/sites/default/files/covid-19-faqs-hospitals-hospital-laboratory-acute-care-facility-data-reporting.pdf.

SPECIFIC COMMENTS

137) Please elaborate this in more detail. Who could vote to in/exclude a certain model's forecast and how?

It was an open forecasting challenge, so modeling teams could choose to not include their models in the ensemble if they were exploratory or otherwise did not want their model included. The only inclusion requirement was that teams submitted their forecast files on-time with the requested formatting. See

inclusion criteria on lines 163-166.

141-142) A “package” cannot generate a forecast. Please state the used method (from the package).

The methodology used in the `simplets` package is described in detail in lines 154-158. Clarifying language, “using the ‘quantile baseline’ method in the `simplets` package with underlying methodology described as follows”, has been added in lines 152-153.

151) Please, state how the ensemble members were casted. E.g. were they paid, if not, what was their motivation to contribute? Did they have the same system knowledge (e.g. modeling team from CDC might have better insights than a hackathon-team from a countryside highschool)?

Participation was voluntary and different teams used different types of data as outlined in the Supplementary Table 1 about data sources. In response to the first question listed, some teams receive direct funding and some don’t, but there was not a monetary incentive to being included in the ensemble.

248) FluSight hub ensemble = FluSight Ensemble? In general, since both FluSight models, the baseline and the ensemble, start with the same prefix (namely FluSight), it is sometimes a little difficult to differentiate between the two when reading. However, since they have fundamentally different roles, it would be good to be able to distinguish between the two quickly and clearly. I would suggest introducing appropriate abbreviations (e.g. BaseFlu, EnsembleFlu) This would make the result section much easier to read.

We thank the reviewer for pointing out the mistake of including “FluSight hub ensemble”, we have updated this instance to “FluSight ensemble”. As far as abbreviating the FluSight baseline and ensemble models, since there are other ensemble models we would still have to distinguish when we are referring to the FluSight ensemble as opposed to the other ensemble models. We prefer to keep our original nomenclature that was used throughout the duration of the study and historically through previous influenza forecasting challenges in the United States.

387) I guess you want to refer to Figure 4 instead?

We have checked and did intend to refer to Figure 3, because we are referring to the state-level relative WIS results that are in Figure 3.

Tables 1 and 2) Having some background information on the used methods in the specific models would drastically increase the informative value of these tables. Currently, it is only some ranked models of which we know nothing about.

We added some classification of model type to Table 1. Further detail on the methods is provided in Supplementary Table 1.

Figure 3) First, in my opinion, this should be a table. Second, there is a lot of content in this table which is not relevant for the manuscript main text. Since the discussion only refers to the spatial heterogeneity,

maybe you find a different way to depict this visually. Finally, the font size is too small, but this is rather a cosmetic comment and you probably know that already.

We thank the reviewer for this comment. We prefer to keep the current version in the main manuscript as the colors highlight our points about consistency of performance across locations for the FluSight Ensemble. We realize however that a table version of this information could be helpful for some readers (for example if they wanted to sort by performance on a particular location). We have opted to provide the information also as a supplementary file and defer to the editors whether it makes sense to include as supplementary data.

References) the link to [8] is broken in the meanwhile

We've added a hyperlink for the referenced PDF.

Reviewer response appendix:

Categorical forecast exploration abstract:

Authors: Rebecca K. Borchering, Jessica Davis, Sarabeth Mathis, et al.

Title: Forecasting categorical trends in influenza hospitalizations: results from an experimental forecasting approach deployed during the 2022-2023 influenza season.

Abstract:

Background and Aims of the Study: During the 2022-2023 influenza season, the U.S. FluSight forecast consortium piloted a new forecasting target of direction and relative magnitude of change in weekly laboratory-confirmed influenza hospital admissions observed over two consecutive weeks. Research in this area could lead to a more robust and timely forecast indicators for use by public health partners, particularly during periods of rapid changes in hospitalizations when the accuracy of forecasts for the absolute levels of new hospitalizations tends to decline.

Methods: Trend definitions of large decrease, decrease, stable, increase, and large increase were defined based, in part, on the distribution of changes in population rates of influenza-associated hospitalizations observed historically in FluSurv-NET (seasons 2010-11 through 2019-20) and HHS-Protect (2021-22 season). Forecasting teams were invited to submit probabilities for each trend category across all fifty states, Washington D.C., and at the national level, weekly, starting December 12, 2022. We compared submitted trend probabilities to probabilities calculated from the quantiles of FluSight Ensemble forecasts for the absolute numbers of hospitalizations. Brier Skill Scores were calculated to assess performance for retrospective trend forecasts based on data available during the 2022-23 influenza season. Submitted trend forecasts were compared to a baseline prediction of equal probability across

trend categories. We also considered differences between an ensemble of the trend submissions and the trend probabilities calculated from the FluSight Ensemble quantile forecasts.

Results: The earliest trends in influenza hospitalizations categorized as increases or large increases were observed in the southeast United States, starting in October 2022. Seven teams submitted trend forecasts based on both statistical and mechanistic forecasting approaches. Methods to calculate trend probabilities varied, but many teams used an approximation of their quantile-based distribution of two-week ahead hospital admission forecasts. Trend forecast performance for team submissions varied by location but generally outperformed the baseline model of equal probabilities.

Implications: Influenza trend forecasts may increase the potential utility, including robustness and timeliness, of forecasts during traditionally difficult times for prediction. Findings from this pilot season of trend forecasts and corresponding output from the FluSight Ensemble may inform directions to improve the real-time usability of influenza forecasts.

REVIEWERS' COMMENTS

Reviewer #1 (Remarks to the Author):

I like the revision, in particular the additional graphical illustrations, the interactive visualization and the improvements on the relative WIS. I only have some minor comments:

1. I would like to see the provided additional national level plots for the forecast weeks ending 11/14/22, 12/5/22, and 2/27/23 as part of Figure 1 or as a new Figure 2 linked to the discussion in the text. In light of the additional plots, it would be great if the same individual model color scheme could be used for (an enlarged version of) Figure 4 providing a link between the WIS and coverage performance of individual models over time. Despite all heterogeneity, it seems that some models were systematically better than others.
2. The geographic heterogeneity is interesting with differently lagged truth situations and the provided additional state level graphics are nice for the supplementary material. It seems that even for a fixed time point individual model performance could vary greatly across states. For the interpretation in Figure 3, it would be good to rather use the space of the uninformative column with the relative WIS of the Flusight baseline instead by the absolute level of the WIS of the baseline across states as absolute reference points.
3. The provided scaling in the color legend in Figure 3 appears to be wrong: in the table, all values below 1 appear blue, all above 1 red.

Reviewer #2 (Remarks to the Author):

My previous comments are somewhat addressed. I don't have additional comments.

Reviewer #3 (Remarks to the Author):

The changes made by the authors have greatly improved the quality of the manuscript. Above all, the manuscript now has - in addition to the already given relevance - the necessary research focus, thanks to the added research question and their treatment in the discussion.

In addition, the authors have also implemented my other comments well. In particular, the classification of the models has greatly improved the information content of the outcome table. As an expert in epidemiological agent-based and differential equation modeling, I am admittedly a little shocked at how small the proportion of mechanistic models is. At the same time it also explains the strong data-focus of the manuscript.

With regard to gaining understanding of ground truth data, I remain convinced that this is also within responsibility of the modeler, as is the mapping of the corresponding uncertainties. The use of internationally agreed and public data sources may satisfy the scientific credo, but it can have devastating consequences if, due to poor data, wrong or biased results nevertheless gain public visibility. Accordingly, I would welcome it if forecasting or scenario hubs generally placed more value and effort on the improvement of data understanding and quality, for example through pure data analysis rounds or the involvement of the actual data collectors. However this is rather a subjective statement from someone knowing both worlds: the innocent beauty of large international data bases and the deep rabbit hole of how awful and biased the corresponding data is actually collected. So it should not influence successful publication of the manuscript - from my point of view, the road is clear.

REVIEWERS' COMMENTS

Reviewer #1 (Remarks to the Author):

I like the revision, in particular the additional graphical illustrations, the interactive visualization and the improvements on the relative WIS. I only have some minor comments:

1. I would like to see the provided additional national level plots for the forecast weeks ending 11/14/22, 12/5/22, and 2/27/23 as part of Figure 1 or as a new Figure 2 linked to the discussion in the text. In light of the additional plots, it would be great if the same individual model color scheme could be used for (an enlarged version of) Figure 4 providing a link between the WIS and coverage performance of individual models over time. Despite all heterogeneity, it seems that some models were systematically better than others. Thank you for this comment, we added the three suggested plots to Figure 1 as panels c, d, and e, with some additional text linking these figures to the discussion in lines 279 and 292. Unfortunately, due to the large number of submitting models, we are unable to update Figure 4 to match the updated colors. We feel this contributes too much distraction from the main messages and it is difficult to distinguish between all of the colors needed to represent the models. We agree that some models performed systematically better than others but believe that this information is best conveyed in Table 1 and in the interactive dashboard where the user can control the number of models that they are viewing to minimize distractions.

2. The geographic heterogeneity is interesting with differently lagged truth situations and the provided additional state level graphics are nice for the supplementary material. It seems that even for a fixed time point individual model performance could vary greatly across states. For the interpretation in Figure 3, it would be good to rather use the space of the uninformative column with the relative WIS of the Flusight baseline instead by the absolute level of the WIS of the baseline across states as absolute reference points. Thank you for this comment. We prefer to keep the relative WIS column for the baseline model as it provides the focal reference point against which the other values should be interpreted. We feel that it is important for the interpretation of the graph to emphasize that the visualization shows how WIS is relative to that of the baseline model. Additionally, the difficulty of interpreting the absolute WIS across locations (due to population expected differences in magnitude) is part of why we do not emphasize the absolute WIS results throughout the manuscript and thus particularly would not want to introduce those values in this figure.

3. The provided scaling in the color legend in Figure 3 appears to be wrong: in the table, all values below 1 appear blue, all above 1 red. Thank you for pointing this out, we corrected the legend as indicated.

Reviewer #2 (Remarks to the Author):

My previous comments are somewhat addressed. I don't have additional comments. Thank you for your reviews.

Reviewer #3 (Remarks to the Author):

The changes made by the authors have greatly improved the quality of the manuscript. Above all, the manuscript now has - in addition to the already given relevance - the necessary research focus, thanks to the added research question and their treatment in the discussion. Thank you!

In addition, the authors have also implemented my other comments well. In particular, the classification of the models has greatly improved the information content of the outcome table. As an expert in epidemiological agent-based and differential equation modeling, I am admittedly a little shocked at how small the proportion of mechanistic models is. At the same time it also explains the strong data-focus of the manuscript.

With regard to gaining understanding of ground truth data, I remain convinced that this is also within responsibility of the modeler, as is the mapping of the corresponding uncertainties. The use of internationally agreed and public data sources may satisfy the scientific credo, but it can have devastating consequences if, due to poor data, wrong or biased results nevertheless gain public visibility. Accordingly, I would welcome it if forecasting or scenario hubs generally placed more value and effort on the improvement of data understanding and quality, for example through pure data analysis rounds or the involvement of the actual data collectors. However this is rather a subjective statement from someone knowing both worlds: the innocent beauty of large international data bases and the deep rabbit hole of how awful and biased the corresponding data is actually collected. So it should not influence successful publication of the manuscript - from my point of view, the road is clear. Thank you, we appreciate the thoughtful nuances of this comment and agree that modelers should take part of the responsibility of verifying the reliability of their data sources. At this time we don't have additions to the manuscript to better describe the data.